# Modeling and therapeutic targeting of inflammation-induced hepatic insulin resistance using human iPSC-derived hepatocytes and macrophages

Marko Groeger [1,2], Koji Matsuo[3,4], Emad Heidary Arash[1,2], Ashley Pereira[3,4], Dounia Le Guillou [5,6], Cindy Pino [6,7], Kayque A. Telles-Silva[2,8], Jacquelyn J. Maher [5,6], Edward C. Hsiao [2,3,4] & Holger Willenbring [1,2,6] ✉

Hepatic insulin resistance is recognized as a driver of type 2 diabetes and fatty liver disease but specific therapies are lacking. Here we explore the potential of human induced pluripotent stem cells (iPSCs) for modeling hepatic insulin resistance in vitro, with a focus on resolving the controversy about the impact of inflammation in the absence of steatosis. For this, we establish the complex insulin signaling cascade and the multiple inter-dependent functions constituting hepatic glucose metabolism in iPSC-derived hepatocytes (iPSC-Heps). Co-culture of these insulin-sensitive iPSC-Heps with isogenic iPSC-derived pro-inflammatory macrophages induces glucose output by preventing insulin from inhibiting gluconeogenesis and glycogenolysis and activating glycolysis. Screening identifies TNFα and IL1β as the mediators of insulin resistance in iPSC-Heps. Neutralizing these cytokines together restores insulin sensitivity in iPSC-Heps more effectively than individual inhibition, reflecting specific effects on insulin signaling and glucose metabolism mediated by NF-κB or JNK. These results show that inflammation is sufficient to induce hepatic insulin resistance and establish a human iPSC-based in vitro model to mechanistically dissect and therapeutically target this metabolic disease driver.

The potential of human induced pluripotent stem cell (iPSC)-derived hepatocytes (iPSC-Heps) for in vitro disease modeling rests on their ability to replicate complex functions provided by the human liver in vivo. One such function that is of high clinical relevance is hepatic glucose metabolism and its regulation by insulin. Some aspects of the insulin signaling cascade have been shown to be active in iPSC-Heps[1,2], but whether these cells allow for the faithful study of disease mechanisms affecting insulin-mediated regulation of glucose metabolism remains to be determined. In fact, it is currently unknown whether insulin and the counterregulatory hormone glucagon exert physiological control over glucose metabolism in iPSC-Heps. Studies are needed that show that the fast-acting and complex hepatic insulin

[1]Division of Transplant Surgery, Department of Surgery, University of California San Francisco, San Francisco, CA 94143, USA. [2]Eli and Edythe Broad Center of Regeneration Medicine and Stem Cell Research, University of California San Francisco, San Francisco, CA 94143, USA. [3]Division of Endocrinology and Metabolism, Department of Medicine, University of California San Francisco, San Francisco, CA 94143, USA. [4]Institute for Human Genetics, University of California San Francisco, San Francisco, CA 94143, USA. [5]Division of Gastroenterology, Department of Medicine, University of California San Francisco, San Francisco, CA 94143, USA. [6]Liver Center, University of California San Francisco, San Francisco, CA 94143, USA. [7]Genomics CoLab, University of California San Francisco, San Francisco, CA 94143, USA. [8]Human Genome and Stem Cell Research Center, University of Sao Paulo, 05508-090 Sao Paulo, Brazil. ✉e-mail: holger.willenbring@ucsf.edu

signaling cascade—revolving around the interaction of the insulin receptor (INSR) with insulin receptor substrate (IRS) 1 and 2 and subsequent activation of the central metabolic regulator protein kinase B (AKT) through phosphoinositide 3-kinase and 3-phosphoinositide-dependent kinase 1[3,4]—regulates gluconeogenesis, glycolysis, glycogen metabolism and thereby glucose output in iPSC-Heps.

iPSC-Heps equipped with fully developed hormone-regulated glucose metabolism would facilitate studies of hepatic insulin resistance, which is common in obesity and critical for manifestation and progression of type 2 diabetes and fatty liver disease[5–7]. Lipid accumulation is thought to be the main cause of hepatic insulin resistance[4]. However, earlier studies in mice showed that inflammation induced by activated macrophages can also impair insulin sensitivity of hepatocytes, leading to disrupted downstream signaling and increased hepatic glucose output[8,9]. Activated macrophages can cause hepatic insulin resistance by secreting pro-inflammatory cytokines[10,11]. In vivo evidence points to interleukin 6 (IL6) as a mediator of the effects activated macrophages have on hepatic glucose metabolism, but comprehensive studies are lacking[12]. Moreover, findings about the contribution of inflammation to hepatic insulin resistance have been contradictory, with some studies showing it acts as an adjuvant to steatosis, whereas others argue it is sufficient by itself [8,9,11,12].

Disruption of hepatic insulin signaling by IL6 has also been shown in vitro using liver cell lines and primary mouse hepatocytes[13]. Similar effects have been reported for tumor necrosis factor α (TNFα) and interleukin 1β (IL1β) in liver cell lines and primary rat hepatocytes[10,14,15]. Moreover, TNFα has been shown in co-cultures of primary rat cells to mediate the disruption of insulin signaling in hepatocytes by activated macrophages[8]. A limitation of these in vitro studies is insufficient analysis of insulin-regulated functions constituting hepatic glucose metabolism, particularly of glucose output, which is at the core of type 2 diabetes and fatty liver disease[16]. In fact, glucose production is reportedly low and not responsive to insulin in liver cell lines[17]. Primary human hepatocytes (PHHs) exhibit insulin-regulated glucose production in vitro as long as differentiation is stabilized by co-culture with stromal cells[18]. Such co-cultures showed that hyperglycemia causes steatosis-associated hepatic insulin resistance[19]. Similarly, spheroid culture maintains insulin sensitivity of PHHs, with signs of insulin resistance developing after long-term metabolic challenge with high-level fatty acids, insulin and glucose[20]. The role of inflammation has not been investigated in these PHH-based models.

Human iPSCs address many limitations of primary cells, including expandability and ability to generate multiple cell types from the same iPSC line, thereby eliminating immunological and genetic biases in co-culture models. iPSC-Heps have been successfully used for human liver disease modeling in vitro; however, incomplete differentiation of iPSC-Heps generated with current protocols requires in-depth characterization to ascertain adequate function[21].

To develop an iPSC-based model of hepatic insulin resistance and resolve the controversy about the impact of inflammation, we modified differentiation of iPSC-Heps to achieve physiologically relevant insulin sensitivity and co-cultured the cells with pro-inflammatory (classically activated; M1) or undifferentiated/non-activated (M0) macrophages derived from the same iPSC line (iPSC-Macs). In addition to defining the effects of inflammation on hepatic insulin signaling and glucose metabolism, we identified the causative cytokines and devised a strategy to restore insulin sensitivity in iPSC-Heps. We ascertained the authenticity of our findings by confirmation in PHHs co-cultured with primary human macrophages (PHMs). These findings establish the potential of our human iPSC-based model for further mechanistic studies and development of new therapies for hepatic insulin resistance.

## Results

### iPSC-Heps exhibit physiological regulation of glucose metabolism

Current protocols for iPSC-Hep generation include insulin and glucocorticoids to enhance differentiation and survival[22–24]. We reasoned that continuous stimulation with these hormones may render iPSC-Heps insensitive to hormone boluses[25,26]. Therefore, we amended our protocol to include a 24-h starvation period without insulin, glucocorticoids or growth factors after the 22-day differentiation process (Supplementary Fig. 1a). We ascertained normal viability (Supplementary Fig. 1b) and differentiation (Supplementary Fig. 1c–e) of iPSC-Heps generated with the amended protocol, including mitochondrial activity (Supplementary Fig. 1f, g).

To determine the insulin sensitivity of our iPSC-Heps, we analyzed glucose production and other functions constituting hepatic glucose metabolism and key aspects of insulin signaling (Fig. 1a). Initially, we observed that an insulin bolus applied after the starvation period was effective in reducing hepatic glucose production but only moderately and briefly (Supplementary Fig. 2a). In contrast, treatment with the physiological insulin antagonist glucagon caused a marked increase in hepatic glucose production for 3 h (Fig. 1b). Increased hepatic glucose production was associated with increased gluconeogenesis, as evidenced by analysis of *PCK1* and *G6PC* gene expression, which was induced by glucagon and reduced by subsequent insulin stimulation (Fig. 1c). *GCK* gene expression indicated that glycolysis was not affected by glucagon but induced by insulin (Fig. 1c). Expression of the three major genes of the INSR complex, *INSR*, *IRS1* and *IRS2*, was reduced by both insulin and glucagon, with maximum reduction after serial stimulation with both hormones, indicating an insulin-dependent negative feedback loop[27,28] (Supplementary Fig. 2b). We excluded that these differences in hepatic glucose production were caused by changes in gene expression of *GLUT2*, the major bi-directional hepatic glucose transporter[29] (Fig. 1c). Prompted by these findings, we added a 1-h glucagon-stimulation period after the 24-h starvation period to our protocol (Supplementary Fig. 1a).

Next, we investigated whether the speed and complexity of insulin-mediated regulation of hepatic glucose production found in primary hepatocytes is replicated by iPSC-Heps[30]. For this, we generated a time course of phosphorylation-mediated activation of INSR and AKT. After insulin bolus, autophosphorylation of INSR and phosphorylation of AKT at T308 and S473 were initiated within 5 min, with AKT phosphorylation increasing for 30 min, indicating involvement of both INSR signaling and mTORC2 signaling, the latter being partially insulin independent[4,31,32]; we also confirmed that insulin causes phosphorylation of p70 S6 kinase (S6K1) at T389, which acts downstream of mTORC1 and is essential for hepatic glucose homeostasis[33] (Supplementary Fig. 2c, Fig. 1d). In addition, we investigated whether glycogen metabolism is regulated by insulin through glycogen phosphorylase L (PYGL)-mediated glycogenolysis and glycogen synthase (GYS)-mediated glycogen synthesis[34]. We found that PYGL is rapidly and progressively dephosphorylated at S15 and thus deactivated after insulin bolus, leading to inhibition of glycogenolysis within 30 min (Fig. 1d). We did not find initiation of glycogen synthesis by activating dephosphorylation of GYS at S641 within 3 h of insulin stimulation (Fig. 1d), although its negative regulator glycogen synthase kinase-3β (GSK3B) appeared to be inactivated by phosphorylation at S9[35] (Supplementary Fig. 2d), probably because glycogen levels were high in iPSC-Heps at baseline (Supplementary Fig. 2e). Nevertheless, the net result of insulin's effects on iPSC-Heps was glycogen accumulation as evidenced by comparison to cells treated with glucagon (Fig. 1e). These results establish that the functions constituting glucose metabolism are fully developed in iPSC-Heps and subject to regulation by insulin and glucagon.

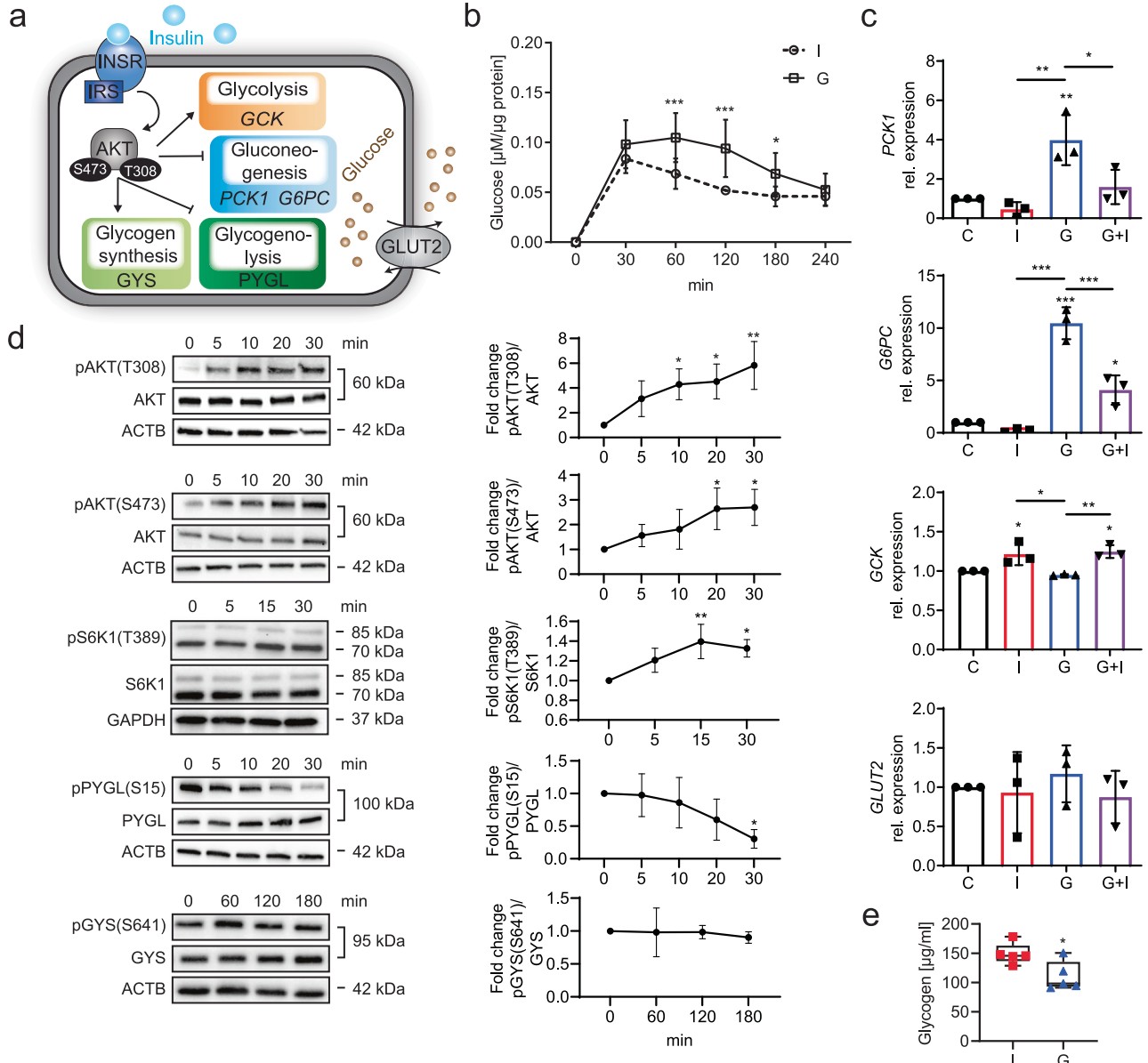

**Fig. 1 | Insulin and glucagon effects on glucose metabolism in iPSC-Heps.**
**a** Overview of insulin signaling and functions constituting glucose metabolism in hepatocytes. **b** Time course of analysis of glucose release into glucose-free media by iPSC-Heps after insulin (I) and glucagon (G). Data are mean ± SD; $n = 6$, two-way ANOVA (two-stage step-up method of Benjamini, Krieger and Yekutieli), *$P < 0.05$ and ***$P < 0.001$ vs. I at indicated timepoints. **c** Gene expression analysis in iPSC-Heps after 4 h of no hormones (C, control), insulin (I), glucagon (G) or 2 h of glucagon followed by 2 h of insulin (G+I). Data are mean ± SD; $n = 3$, one-way

ANOVA (Tukey's test), *$P < 0.05$, **$P < 0.01$ and ***$P < 0.001$ vs. C or between indicated conditions. **d** Time course of western blot analysis of AKT, S6K1, PYGL and GYS phosphorylation in iPSC-Heps after insulin. Data are mean ± SD; $n = 3$, one-way ANOVA (Dunnett's test), *$P < 0.05$ and **$P < 0.01$ vs. time point 0 min.
**e** Quantification of glycogen in iPSC-Heps after 3 h of insulin or glucagon. Data are mean ± SD; $n = 5$, unpaired two-tailed Student's t test, *$P < 0.05$. Source data are provided as a Source Data file.

## M1 iPSC-Macs cause inflammation and insulin resistance in iPSC-Heps

Previous studies used iPSCs to model fatty liver disease, including macrophage-mediated inflammation, but they focused on lipid metabolism and fibrosis[36,37]. To investigate how macrophage-mediated inflammation affects glucose metabolism in iPSC-Heps, we co-cultured them with M1 or M0 iPSC-Macs generated from the same healthy-donor iPSC line (WTC[38]) using our recently published differentiation protocol[39] (Supplementary Fig. 1a). Because macrophages are thought to disrupt glucose metabolism by secreting pro-inflammatory cytokines[40], we established indirect 24-h co-culture of iPSC-Heps with M1 or M0 iPSC-Macs in a cell-impermeable transwell system (Fig. 2a). Before co-culture we confirmed that iPSC-Macs are viable and

express principal markers of the hematopoietic and monocytic lineages as well as activation-specific markers (Supplementary Fig. 3a, b).

First, we assessed manifestation of inflammation of iPSC-Heps, which showed that co-culture with M1 iPSC-Macs, but not with M0 iPSC-Macs or mono-culture, causes damage and death, as evidenced by measurement of lactate dehydrogenase (LDH) in the media and cleaved caspase 3 immunofluorescence (Fig. 2b, Supplementary Fig. 3c). Analysis of *CASP1*, *CASP4* and *CASP5* gene expression pointed to pyroptosis as a mechanism involved in iPSC-Hep death, which accords with previous findings in mice[41] (Supplementary Fig. 3d). Moreover, we found activation of the NF-κB and JNK signaling pathways, which are known to play prominent roles in inflammation-induced hepatocyte damage and hepatic insulin resistance[42].

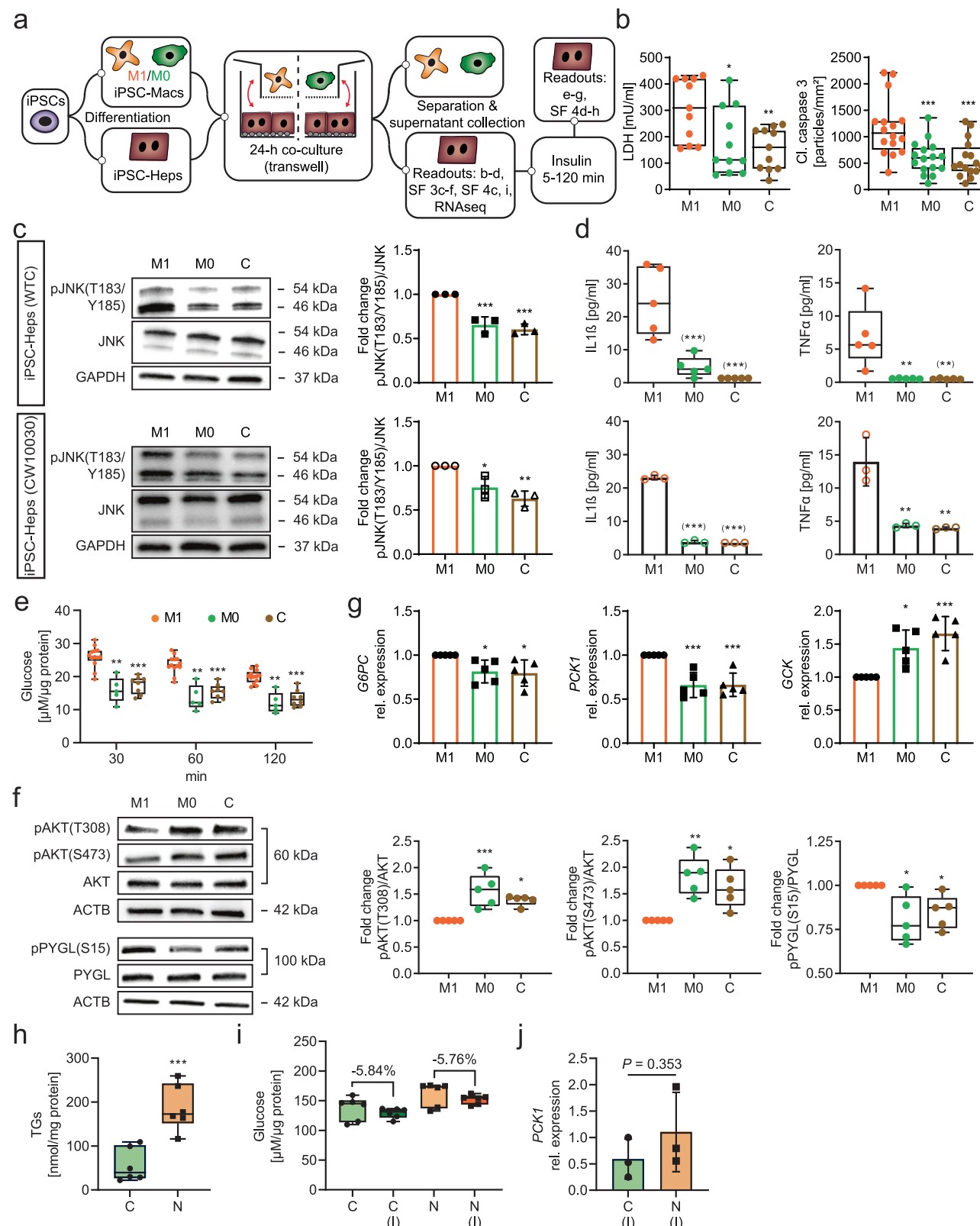

Specifically, in addition to increased expression of the marker genes *NFKB2* and *TNF*, we found increased phosphorylation of JNK at T183/Y185 in iPSC-Heps co-cultured with M1 iPSC-Macs; we confirmed these results in another healthy-donor iPSC line (CW10030[43]) (Fig. 2c, Supplementary Fig. 3e). In accord, pro-inflammatory cytokines such as IL1β, TNFα, IL6 and interferon γ (IFNγ) were much more abundant in media from co-cultures of iPSC-Heps with M1 than with M0 iPSC-Macs

or from iPSC-Hep mono-cultures, despite a mitigating effect of iPSC-Heps on cytokine and chemokine release by M1 iPSC-Macs (Fig. 2d, Supplementary Fig. 3f).

Next, we asked whether insulin sensitivity and functions constituting glucose metabolism are affected in iPSC-Heps by iPSC-Mac co-culture. Indeed, after insulin bolus we found higher glucose levels in media from co-cultures of iPSC-Heps with M1 than with M0 iPSC-Macs

**Fig. 2 | Inflammation and glucose metabolism changes in iPSC-Heps co-cultured with M1 or M0 iPSC-Macs or accumulating lipid. a** Overview of experimental approach. SF, Supplementary Fig. **b** Quantification of LDH in media ($n = 11$) and of immunofluorescence of cleaved caspase 3 particles in iPSC-Heps ($n = 4$, 3 random regions/$n$). Data are mean ± SD; one-way ANOVA (Dunnett's test), *$P < 0.05$, **$P < 0.01$ and ***$P < 0.001$ vs. M1. **c, d** Western blot analysis of JNK phosphorylation in iPSC-Heps ($n = 3$) (**c**) and pro-inflammatory cytokine release into media (WTC: $n = 5$, CW10030: $n = 3$) (**d**) after 24-h co-culture of iPSC-Heps with M1 or M0 iPSC-Macs or in iPSC-Hep mono-culture (C, control) comparing cells generated from the WTC and CW10030 iPSC lines. Data are mean ± SD; one-way ANOVA (Dunnett's test), *$P < 0.05$, **$P < 0.01$ and ***$P < 0.001$ vs. M1. Asterisks in parentheses indicate values set to the detection limit of the assay. **e–g** Time course of analysis of glucose release into 1 mM glucose-containing media after insulin (M1: $n = 13$, M0: $n = 5$, C: $n = 8$) (**e**), western blot analysis of AKT and PYGL phosphorylation in iPSC-Heps 30 min after insulin ($n = 5$) (**f**) and gene expression analysis in iPSC-Heps 1 h after insulin ($n = 5$) (**g**) after 24-h co-culture of iPSC-Heps with M1 or M0 iPSC-Macs or in iPSC-Hep mono-culture (C, control). Data are mean ± SD; two-way (**e**) or one-way (**f, g**) ANOVA (Dunnett's test), *$P < 0.05$, **$P < 0.01$ and ***$P < 0.001$ vs. M1. **h–j** Quantification of triglycerides (TGs) in iPSC-Heps ($n = 6$) (**h**), quantification of glucose release into 1 mM glucose-containing media after 2 h without or with insulin (I) ($n = 6$) (**i**) and gene expression analysis in iPSC-Heps 1 h after insulin ($n = 3$) (**j**) comparing iPSC-Heps from 3 healthy (C, control) and 3 NASH (N) patients. Data are mean ± SD; unpaired two-tailed Student's t test, ***$P < 0.001$. Source data are provided as a Source Data file.

or from iPSC-Hep mono-cultures (Fig. 2e). In contrast to M1 iPSC-Mac co-culture, 24-h treatment with any one of the pro-inflammatory cytokines IL1β, TNFα, IL6 or IFNγ—at a dose reportedly causing damage and impaired insulin signaling in primary hepatocytes[13–15,44]—failed to increase LDH or post-insulin glucose levels in media from iPSC-Hep mono-cultures (Supplementary Fig. 4a, b). Thus, iPSC-Heps mimic in vitro the resilience characteristic for primary hepatocytes in vivo[45–47], which excludes that M1 iPSC-Mac-induced insulin resistance of iPSC-Heps is a mere byproduct of damage or death. Moreover, these results highlight that the complexity of the interaction of M1 iPSC-Macs and iPSC-Heps cannot be faithfully replicated with single cytokines.

To determine how M1 iPSC-Macs disrupt insulin-mediated regulation of hepatic glucose production, we compared co-cultures of M1 or M0 iPSC-Macs with iPSC-Heps and iPSC-Hep mono-cultures using the readouts described in Fig. 1a. At baseline, we found no effect of iPSC-Mac co-culture on glycogen levels in iPSC-Heps (Supplementary Fig. 4c). After insulin bolus, we found increased *IRS1* and *IRS2* gene expression in iPSC-Heps co-cultured with M1 iPSC-Macs, indicating that the negative feedback loop observed in iPSC-Hep mono-cultures was disrupted; *GLUT2* gene expression was not altered (Supplementary Figs. 4d, 2b). Moreover, inhibitory phosphorylation of IRS1 at S307 was increased in these cells, which has a negative effect on insulin signaling[3] (Supplementary Fig. 4e). In accord, insulin-induced activation of AKT by phosphorylation at T308 and S473 was impaired (Fig. 2f), resulting in reduced activation of S6K1 by phosphorylation at T389 (Supplementary Fig. 4f). At the functional level, we found failure to downregulate *G6PC* and *PCK1* and upregulate *GCK* gene expression in response to insulin, leading to increased gluconeogenesis and decreased glycolysis (Fig. 2g). In addition, glycogenolysis continued to be active despite insulin bolus, as evidenced by lack of dephosphorylation of PYGL at S15 (Fig. 2f). The effect of insulin on the phosphorylation of GYS at S641 in iPSC-Heps was similar among all culture conditions, excluding a substantial contribution of impaired glycogen synthesis to increased hepatic glucose production caused by M1 iPSC-Macs (Supplementary Fig. 4g). Gene expression of the lipogenic transcription factor SREBP-1c was decreased in iPSC-Heps co-cultured with M1 iPSC-Macs (Supplementary Fig. 4h), which further illustrates impaired AKT activation and excludes de novo lipogenesis as the cause of insulin resistance[48,49]. In accord, triglyceride staining showed no lipid accumulation in iPSC-Heps co-cultured with M1 iPSC-Macs (Supplementary Fig. 4i).

Prompted by the rapid onset of M1 iPSC-Mac-induced insulin resistance, we investigated whether lipid accumulation is similarly disruptive in iPSC-Heps using gluconeogenesis and glycolysis as readouts. We found that treatment with the fatty acids oleate and palmitate for 6 days caused steatosis but had no effect on insulin-mediated regulation of glucose production and *PCK1* and *GCK* gene expression in iPSC-Heps in mono-culture (Supplementary Fig. 4j–l). We confirmed these results in iPSC-Heps genetically predisposed to lipid accumulation[50]. For this, we used iPSC lines from three donors homozygous for the PNPLA3 I148M variant who have biopsy-confirmed nonalcoholic steatohepatitis (NASH)[43]. Comparison to iPSC-Heps generated from three healthy-donor iPSC lines[43] showed increased triglycerides in the NASH iPSC-Heps but normal insulin sensitivity as evidenced by unaltered glucose output and *PCK1* and *SREBP1c* gene expression after insulin bolus (Fig. 2h–j, Supplementary Fig. 4m). In addition, gene expression of *JUN*, a transcription factor activated by JNK under lipotoxic conditions[51], was unaltered (Supplementary Fig. 4n). These results show that M1 iPSC-Macs, but not steatosis, rapidly cause insulin resistance in iPSC-Heps by secreting one or more soluble factors—probably pro-inflammatory cytokines—that activate NF-κB and JNK signaling and thereby inhibit IRS-mediated activation of AKT[52,53].

## TNFα, IL1β and IFNγ mediate M1 iPSC-Mac-induced inflammation of iPSC-Heps

To identify the M1 iPSC-Mac-derived soluble factors responsible for causing inflammation and insulin resistance in iPSC-Heps, we investigated whether specific pathways were activated by RNA sequencing (RNAseq). Bioinformatic analysis revealed high similarity between iPSC-Hep mono-cultures and co-cultures with M0 iPSC-Macs, prompting us to focus on the 2937 genes significantly differentially expressed between iPSC-Heps co-cultured with M1 or M0 iPSC-Macs (Fig. 3a–c, Supplementary Data 1).

Functional annotation using Database for Annotation, Visualization and Integrated Discovery (DAVID) showed that many gene clusters related to inflammation were upregulated in iPSC-Heps co-cultured with M1 iPSC-Macs, with NF-κB, TNF, IL1 and IFN signaling being most significantly induced (Fig. 3d). In addition, *CXCL2*, *CXCL9*, *CX3CL1* and *CCL20*, genes involved in chemotaxis and immune cell recruitment, were also upregulated (Supplementary Fig. 5a). In contrast, metabolic and synthetic functions were downregulated in iPSC-Heps co-cultured with M1 iPSC-Macs (Supplementary Fig. 5b).

To further substantiate these results, we investigated whether the observed gene expression changes could be linked to extracellular regulators using Ingenuity Pathway Analysis (IPA). This analysis confirmed TNFα, IL1β and IFNγ as the pro-inflammatory cytokines with the most profound effects on iPSC-Heps co-cultured with M1 iPSC-Macs (Fig. 3e). Moreover, IPA suggested that IL1β exerts its pro-inflammatory effects by activating the *NFKB1* and *IRF1* genes (Fig. 3e). These results show the prominent role of TNFα, IL1β and IFNγ in M1 iPSC-Mac-induced inflammation of iPSC-Heps.

## Neutralizing M1 iPSC-Mac-derived TNFα and IL1β restores insulin sensitivity in iPSC-Heps

TNFα and IL1β have been independently pursued as therapeutic targets in clinical trials for type 2 diabetes and systemic insulin resistance[54–56]. To determine the contribution of these pro-inflammatory cytokines to M1 iPSC-Mac-induced insulin resistance in iPSC-Heps, we inhibited them individually or in combination in co-cultures of iPSC-Heps with M1 iPSC-Macs using neutralizing antibodies. Media LDH levels and *CASP1* gene expression indicated iPSC-Heps were not protected from damage or death by TNFα and/or IL1β

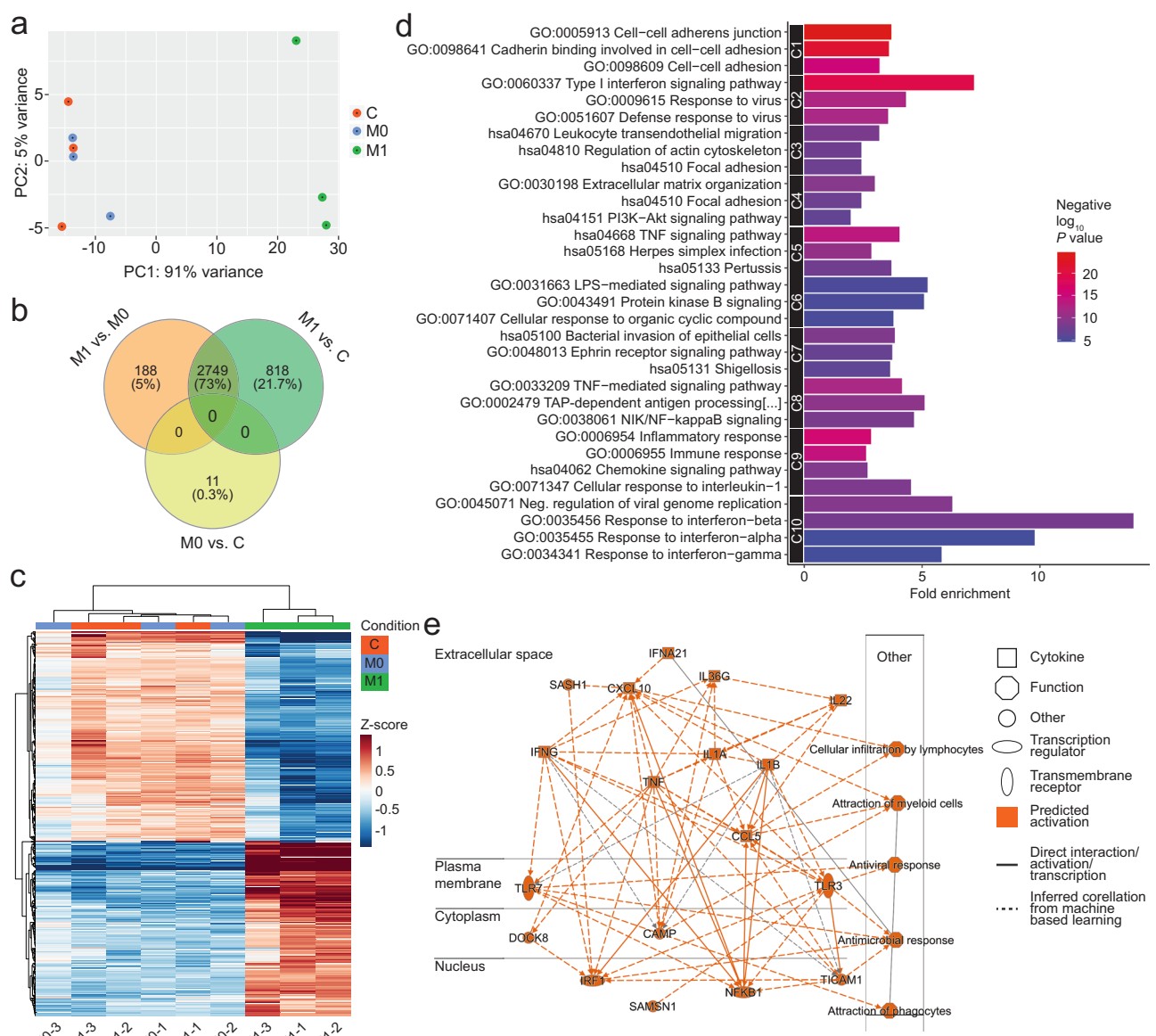

**Fig. 3 | RNAseq analysis of iPSC-Heps co-cultured with M1 or M0 iPSC-Macs.**
**a** Principal component analysis of gene expression profile of iPSC-Heps after 24-h co-culture with M1 or M0 iPSC-Macs or in iPSC-Hep mono-culture (C, control). $n = 3$.
**b** Venn diagram of significantly differentially expressed genes comparing the indicated conditions. $n = 3$, FDR-adjusted $P$ value ($P < 0.05$) by Wald test. **c** Heatmap of the top 1000 differentially expressed genes in iPSC-Heps under the indicated conditions in three independent experiments. The Z-score represents the gene-wise deviation from the mean of the log-transformed and variance-stabilized read counts. $n = 3$. **d** Top 10 upregulated pathway clusters enriched (Cluster Enrichment

Score > 1.3) in the genes differentially expressed between iPSC Heps co-cultured with M1 or M0 iPSC-Macs identified using DAVID. Vertical axis represents enrichment fold values and horizontal axis shows the names of GO-BP, GO-MF and GO-CC terms and KEGG pathways. Node color indicates the enrichment significance, red represents higher significance. $n = 3$, FDR-adjusted $P$ value ($P < 0.05$) by Wald test.
**e** Molecular activity predictor pathway analysis generated using IPA representing the regulatory effects with top consistency scores showing TNFα, IL1β and IFNγ as the pro-inflammatory cytokines most active on iPSC-Heps co-cultured with M1 iPSC-Macs. $n = 3$, FDR-adjusted $P$ value ($P < 0.05$) by Wald test.

neutralization (Fig. 4a, Supplementary Fig. 6a). However, NF-κB and JNK signaling was mitigated by neutralization of TNFα alone or in combination with IL1β, as evidenced by reduction of *NFKB2* or *TNF* gene expression, respectively (Fig. 4b, Supplementary Fig. 6a). Neutralization of IL1β alone or in combination with TNFα was effective in reducing activating phosphorylation of JNK at T183/Y185 (Fig. 4c). Interestingly, TNFα neutralization also impacted M1 iPSC-Macs, not their principal polarization but susceptibility to recruitment as evidenced by analysis of CD86 and CCR2 cell surface levels[57] (Supplementary Fig. 6b).

Next, we investigated the effect of TNFα and/or IL1β neutralization on insulin sensitivity and functions constituting glucose metabolism in iPSC-Heps co-cultured with M1 iPSC-Macs. We found

that media glucose levels were reduced by IL1β neutralization, alone or in combination with TNFα, at 30 min and for 2 h after insulin bolus; in contrast, neutralization of TNFα alone showed an effect only after 2 h (Fig. 4d). At the signaling level, TNFα and/or IL1β neutralization only partially restored the insulin-dependent negative feedback loop regulating the INSR complex, as evidenced by suppression of gene expression of *IRS2*, but not *IRS1*, and unaltered phosphorylation of IRS1 at S307 (Supplementary Fig. 6c, d). However, neutralization of both TNFα and IL1β restored activating phosphorylation of AKT at T308 and S473 (Fig. 4e). At the functional level, we found that neutralization of TNFα alone or in combination with IL1β promoted insulin-induced reduction of gene expression of *G6PC* but not *PCK1*, which required neutralization of both TNFα and IL1β; *GCK* gene

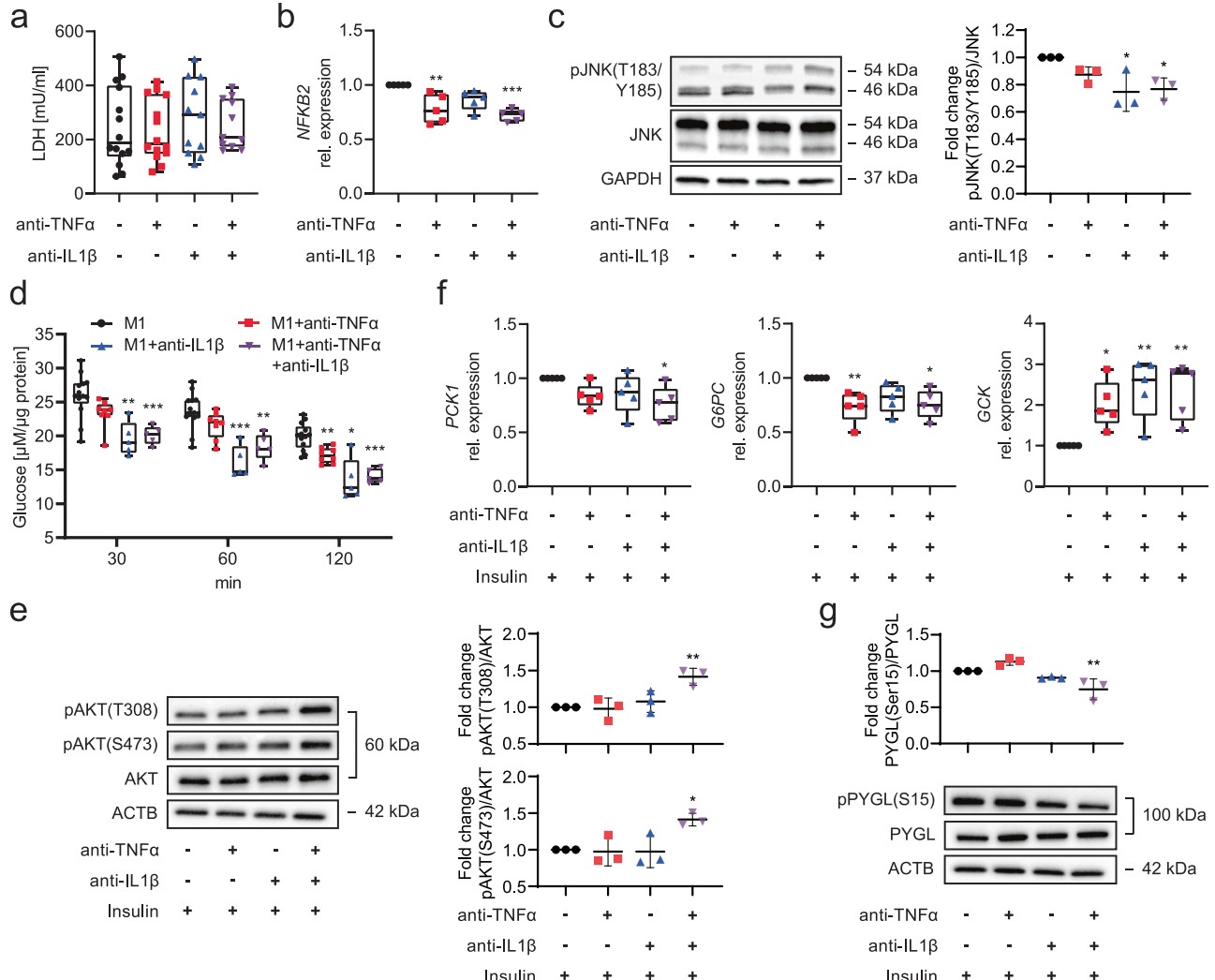

**Fig. 4 | Effect of TNFα and/or IL1β neutralization on inflammation and glucose metabolism changes in iPSC-Heps co-cultured with M1 iPSC-Macs.** **a–c** Quantification of LDH in media (M1 and M1+anti-TNFα: $n = 14$, M1+anti-IL1β and M1+anti-TNFα+anti-IL1β: $n = 11$) (**a**), gene expression analysis in iPSC-Heps ($n = 5$) (**b**) and western blot analysis of JNK phosphorylation in iPSC-Heps ($n = 3$) (**c**) after 24-h co-culture of iPSC-Heps with M1 iPSC-Macs and indicated antibody treatments. Data are mean ± SD; one-way ANOVA (Dunnett's test), *$P < 0.05$, **$P < 0.01$ and ***$P < 0.001$ vs. no-antibody condition. **d–g** Time course of analysis of glucose release into 1 mM glucose-containing media after insulin (M1: $n = 13$, M1+anti-TNFα:

$n = 8$, M1+anti-IL1β and M1+anti-TNFα+anti-IL1β: $n = 5$) (**d**), western blot analysis of AKT phosphorylation in iPSC-Heps 30 min after insulin ($n = 3$) (**e**), gene expression analysis in iPSC-Heps 1 h after insulin ($n = 5$) (**f**) and western blot analysis of PYGL phosphorylation in iPSC-Heps 30 min after insulin ($n = 3$) (**g**) after 24-h co-culture of iPSC-Heps with M1 iPSC-Macs and indicated antibody treatments. Data are mean ± SD; two-way (**d**) and one-way (**e–g**) ANOVA (Dunnett's test), *$P < 0.05$, **$P < 0.01$ and ***$P < 0.001$ vs. no-antibody condition. Source data are provided as a Source Data file.

expression was increased after neutralization of TNFα and/or IL1β (Fig. 4f). Neutralization of both TNFα and IL1β caused deactivating dephosphorylation of PYGL at S15 (Fig. 4g). These results show that M1 iPSC-Macs disrupt insulin-mediated regulation of gluconeogenesis, glycolysis and glycogenolysis in iPSC-Heps by secreting TNFα and IL1β and that both must be neutralized to restore insulin sensitivity to near normal.

## TNFα and IL1β neutralization reverses inflammation-induced insulin resistance in PHHs

Finally, we determined whether our iPSC-based model accurately reflects primary cells. We found a similar pattern of increased IL1β and TNFα media levels in 24-h co-cultures of PHHs with M1 PHMs, with more contribution from PHHs than observed for iPSC-Heps (Figs. 5a, 2d). Because we could not detect differences in phosphorylation of JNK at T183/Y185 between PHHs co-cultured with M1 or M0 PHMs, or PHH mono-cultures, at this time point (Fig. 5b),

we generated a time course, which showed that this modification of JNK already occurred in PHHs after 2 h of co-culture with M1 PHMs (Supplementary Fig. 7a). While JNK phosphorylation subsequently declined, gene expression of the effector of JNK-mediated inflammation *JUN* remained increased in PHHs after 24 h of co-culture with M1 PHMs (Fig. 5c). Combined antibody-mediated neutralization of IL1β and TNFα was effective in reversing increased gene expression of *JUN*, *NFKB2* and *TNF* but not *CASP1* at this time point (Fig. 5c), exactly as in our iPSC-based model (Fig. 4b, Supplementary Fig. 6a).

We also investigated inflammation-induced insulin resistance in co-cultures of PHHs and PHMs. We performed these experiments side by side with iPSC-Heps and iPSC-Macs generated from the CW10030 line to facilitate direct comparison and validate our results obtained with the WTC iPSC line. For both primary cells and iPSC derivatives, after insulin bolus we found higher media glucose levels, impaired activation of AKT by phosphorylation at S473 and failure to

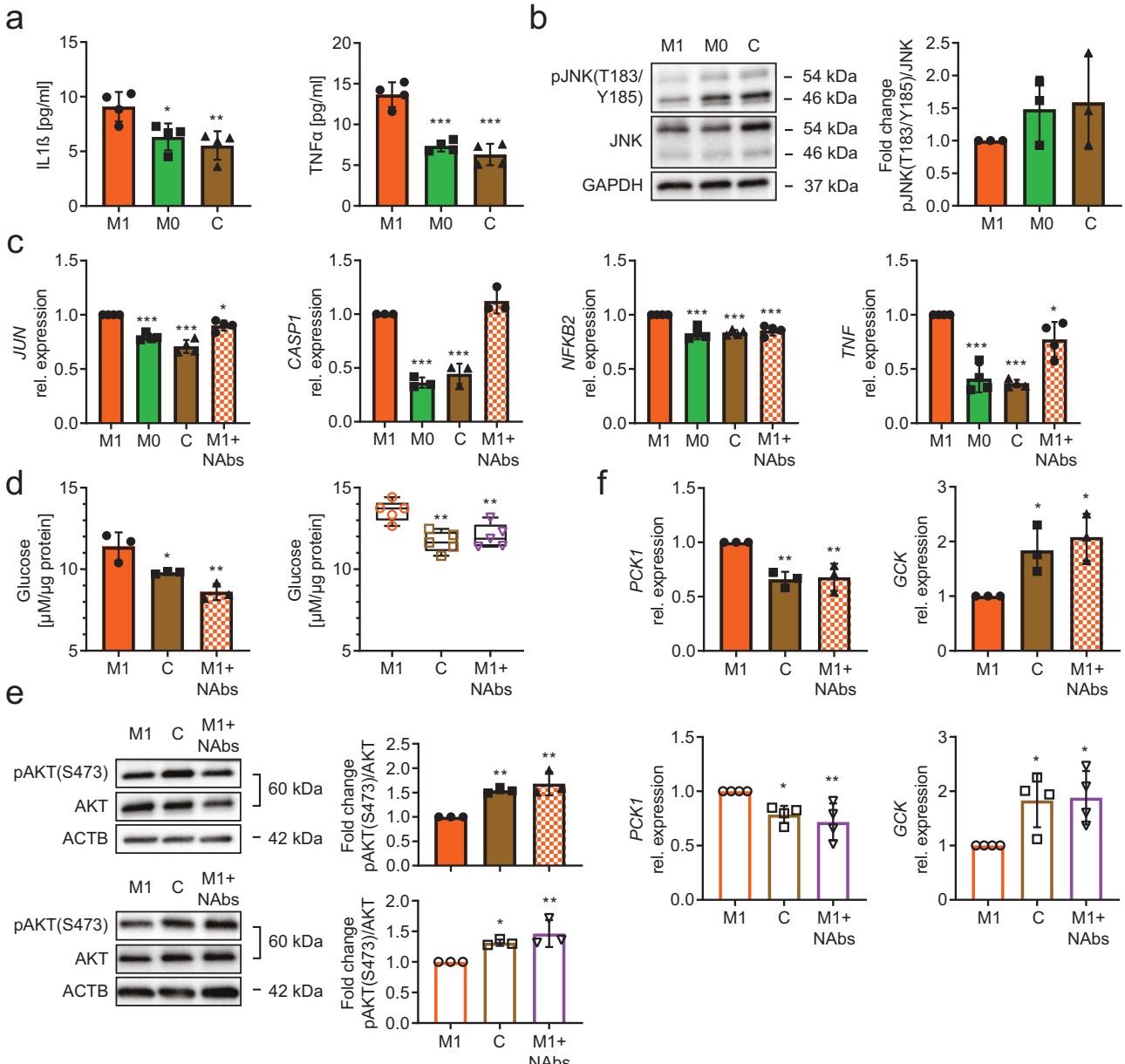

**Fig. 5 | Effect of TNFα and/or IL1β neutralization on inflammation and glucose metabolism changes in PHHs co-cultured with M1 PHMs. a, b** Pro-inflammatory cytokine release into media (*n* = 4) (**a**) and western blot analysis of JNK phosphorylation (*n* = 3) (**b**) in PHHs after 24-h co-culture with M1 or M0 PHMs or in PHH mono-culture (C, control). Data are mean ± SD; one-way ANOVA (Dunnett's test), \**P* < 0.05, \*\**P* < 0.01 and \*\*\**P* < 0.001 vs. M1. **c** Gene expression analysis of PHHs after 24-h co-culture with M1 or M0 PHMs, in PHH mono-culture (C, control) or after 24-h co-culture with M1 PHMs including TNFα and IL1β neutralizing antibodies (M1+NAbs). Data are mean ± SD; *JUN, NFKB2, TNF*: *n* = 4, *CASP1*: *n* = 3, one-way ANOVA (Dunnett's test), \**P* < 0.05 and \*\*\**P* < 0.001 vs. M1. **d–f** Analysis of glucose

release into 1 mM glucose-containing media 2 h after insulin (primary cells: *n* = 3, CW10030 iPSC-derived cells: *n* = 5) (**d**), western blot analysis of AKT phosphorylation 30 min after insulin (*n* = 3) (**e**) and gene expression analysis 1 h after insulin (primary cells: *n* = 3, CW10030 iPSC-derived cells: *n* = 4) (**f**) in hepatocytes after 24-h co-culture with M1 macrophages or in hepatocyte mono-culture (C, control) or after 24-h co-culture with M1 macrophages including TNFα and IL1β neutralizing antibodies (M1+NAbs). Primary cells, filled bars/symbols; CW10030 iPSC-derived cells, open bars/symbols. Data are mean ± SD; one-way ANOVA (Dunnett's test), \**P* < 0.05 and \*\**P* < 0.01 vs. M1. Source data are provided as a Source Data file.

downregulate *PCK1* and upregulate *GCK* gene expression in co-cultures of hepatocytes with M1 macrophages, indicating insulin resistance in hepatocytes, which could be reversed by neutralization of IL1β and TNFα (Fig. 5d–f). We also confirmed in primary cells our finding made in iPSC derivatives that M1 macrophages induce gene expression of *G6PC* in hepatocytes, including IL1β and TNFα acting as mediators (Supplementary Fig. 7b). Primary-cell co-cultures differed from co-cultures of derivatives of both iPSC lines in that insulin dephosphorylated PYGL at S15 in PHHs normally whereas phosphorylation of GSK3B at S9 was impaired in PHHs by IL1β and TNFα released by M1

PHMs (Supplementary Fig. 7c, d), resulting in decreased glycogenolysis and increased glycogen synthesis, probably as a result of much lower glycogen levels in PHHs than in iPSC-Heps (Supplementary Fig. 2e).

## Discussion

Here we developed an iPSC-based model of hepatic insulin resistance, a common clinical condition that drives type 2 diabetes and fatty liver disease in obesity. For this, we generated iPSC-Heps with hormone-regulated glucose metabolism and, to define the impact of inflammation, established co-culture with isogenic M1 or M0 iPSC-Macs. We

ascertained the authenticity of our iPSC-based model by comparison to co-cultures of PHHs with PHMs.

Specifically, our iPSC-Heps replicate the physiological hepatic glucose production identified in clinical studies[58], including rapid response to insulin and glucagon boluses. Our iPSC-Heps also replicate the mechanistic underpinnings, that is, the complex hepatic insulin signaling cascade regulating gluconeogenesis, glycolysis and glycogen metabolism, which could thus far only be studied in animal models and primary hepatocytes[17,59]. In addition, as reported for hepatocytes in patients with fatty liver disease[60], we found that our iPSC-Heps undergo inflammation-induced damage and death (possibly pyroptosis) when exposed to M1 iPSC-Macs. Moreover, our iPSC-Heps respond to M1 iPSC-Mac co-culture by downregulating metabolic and synthetic functions, which supports clinical data suggesting hepatocyte dedifferentiation as a driver of fatty liver disease[61,62].

Taking advantage of the responsiveness of our iPSC-Heps to co-culture with M1 iPSC-Macs, we investigated whether macrophage-mediated inflammation is sufficient to induce hepatic insulin resistance. Indeed, we found increased glucose output by iPSC-Heps co-cultured with M1 iPSC-Macs in the absence of steatosis. Moreover, we found that iPSC-Heps remain insulin sensitive in two models of steatosis: exogenous fatty acid challenge and genetic predisposition caused by the PNPLA3 I148M variant[50]. The latter finding is consistent with the clinical observation that PNPLA3 I148M-associated steatosis is not linked to hepatic insulin resistance[50,63]. These findings highlight the rapid and profound effect of macrophage-mediated inflammation on insulin signaling and glucose metabolism in hepatocytes and question whether lipid accumulation alone causes insulin resistance[4]. In accord, reports that steatosis-induced hepatic insulin resistance coincides with lipotoxicity and ER stress suggest involvement of some of the same inflammatory signaling pathways as induced by macrophages, particularly JNK[52,64,65].

At the mechanistic level, we found that multiple dysfunctions—increased gluconeogenesis and glycogenolysis and decreased glycolysis—contribute to M1 iPSC-Mac-induced insulin resistance in iPSC-Heps, which accords with original observations in patients with hepatic insulin resistance in the setting of type 2 diabetes[66-68] but is at odds with findings in animal models that the acute action of insulin in hepatocytes is limited to glycogen metabolism[34,69]. In fact, we found that insulin has a strong effect on glycogenolysis but none on glycogen synthesis in iPSC-Hep mono-cultures. However, this finding is probably due to high glycogen levels in iPSC-Heps leading to suppression of glycogen synthesis because insulin's effects in co-cultures of M1 PHMs with PHHs with low glycogen levels were reversed[70], which warrants further studies of the regulation of the enzymes involved in glycogen homeostasis, including the effects of allosteric regulation[71].

The authentic interplay between our iPSC-Heps and M1 iPSC-Macs allowed us to identify TNFα and IL1β as the cytokines causing inflammation and insulin resistance in iPSC-Heps. Neutralization of these cytokines revealed specificity in how they cause inflammation, with TNFα acting through NF-κB and IL1β through JNK, both pathways implicated in hepatic insulin resistance[52,53]. These cytokines also differed in effect on insulin-regulated glucose production by iPSC-Heps, with TNFα acting slower and weaker than IL1β. These findings suggest that decreased glycolysis plays a major role in inflammation-induced insulin resistance in iPSC-Heps because glycolysis has been shown in mice to be more affected by IL1β than TNFα[72].

In accord with different mechanisms of action, neutralization of both TNFα and IL1β showed an additive effect on insulin resistance in iPSC-Heps, which we confirmed in PHHs co-cultured with PHMs. Together, these findings provide an explanation for mixed outcomes of clinical trials in which either of these cytokines was inhibited to restore systemic insulin sensitivity in patients with type 2 diabetes[56].

Our finding that single TNFα or IL1β boluses failed to increase glucose output by iPSC-Heps shows that faithful modeling of macrophage-induced inflammation of hepatocytes requires sustained secretion and interaction of multiple cytokines as provided by iPSC-Mac co-culture. By using iPSC-Heps and iPSC-Macs generated from the same iPSC line our model facilitates bias-free validation of genetic risk factors and screens for new therapeutic targets. Along these lines, our model could readily be expanded to include gene editing and other iPSC-derived cell types to study the effects of hepatic insulin resistance on metabolic disease progression in and outside the liver.

In summary, our results establish the efficacy of iPSC-Hep and iPSC-Mac co-culture as an authentic and tunable model of inflammation-induced hepatic insulin resistance that facilitates in-depth mechanistic studies and development of new therapies.

## Methods

### Experimental model and iPSC culture
The human male healthy-donor iPSC line GM25256 (WTC[38]) was used for most experiments. In addition, the healthy-donor iPSC lines CW10001, CW10030 and CW10037 and the NASH-patient iPSC lines CW10152, CW10201 and CW10208 from the CIRM iPSC Repository available at Fujifilm Cellular Dynamics were used[43]. Undifferentiated iPSCs were cultured in mTeSR1 (StemCell Technologies) in six-well plates coated with Cultrex Reduced Growth Factor Basement Membrane Extract (RGF-BME; R&D Systems) at 1:30 dilution in Knockout DMEM (Gibco) at 37 °C in 5% $CO_2$ and 5% $O_2$.

### Differentiation of iPSC-Heps
Differentiation was performed at 37 °C in 5% $CO_2$ and 5% $O_2$ unless stated otherwise. iPSCs were differentiated into iPSC-Heps using a refined version of a previously published protocol[73]. At 70 to 80% confluency, colonies of iPSCs were detached using Accutase (StemCell Technologies) and 100,000 cells/cm² diluted in mTeSR1 including 10 µM Y-27632 (Thermo Fisher Scientific) were seeded in RGF-BME-coated six-well plates. After 24 h, medium was changed to endoderm-induction medium (EIM), consisting of RPMI 1640 (Life Technologies) containing 2% Gem21 without insulin (GeminiBio), 1% Glutamax (Gibco), 1% non-essential amino acids solution (NEAA; Gibco), 0.5 mM sodium butyrate (Sigma-Aldrich) and 100 ng/ml activin A (StemCell Technologies), for 7 days in 20% $O_2$. The following compounds were added to EIM during the first 3 days: 3 µM CHIR (StemCell Technologies) on day 1, 20 ng/ml basic fibroblast growth factor (FGFb; Peprotech) and 10 ng/ml bone morphogenetic protein 4 (BMP4; Peprotech) on days 1 and 2, 50 nM PI103 (Thermo Fisher Scientific) on days 1 to 3, knockout serum replacement (KSR; Life Technologies) at 2% on day 1, 1% on day 2 and 0.2% on day 3.

On days 8 to 17, cells were cultured in hepatic induction medium (HIM), consisting of IMDM (Thermo Fisher Scientific) containing 1% Glutamax, 1% NEAA, 100 nM dexamethasone (Sigma-Aldrich), 100 nM insulin (GeminiBio) and 0.5 mM 1-thioglycerol (Sigma-Aldrich). The following compounds were added to HIM between days 8 and 17: 10 ng/ml FGFb and 20 ng/ml BMP4 between days 8 and 17, 20 ng/ml hepatocyte growth factor (HGF; Peprotech) between days 12 and 17. On day 10, cells were detached using 0.25% trypsin-EDTA (Gibco) and split 1:2 into RGF-BME-coated 12 or 24-well plates.

On days 18 to 22, cells were cultured in Hepatocyte Culture Medium BulletKit (HCM; Lonza) without epidermal growth factor, including 20 ng/ml oncostatin M (Peprotech) and 20 ng/ml HGF in 20% $O_2$. After day 22 (end of differentiation), iPSC-Heps were used within 7 days for experiments. Medium was changed daily during differentiation and maintenance.

### PHH culture
Cryopreserved, plateable PHHs (Lot: BMO) were purchased from BioIVT; the cells were reported to be metabolically active and isolated from a 45-year-old Caucasian male with BMI of 22.6 and no history of

excessive alcohol consumption or smoking. PHHs were plated at a density of 250,000 viable cells/cm² in collagen I-coated 24-well plates (Corning) in INVITROGRO CP medium (BioIVT). After attachment, media was changed to HCM including 20 ng/ml HGF at 37 °C in 5% $CO_2$ and 5% $O_2$. After 24 h, PHHs were used for co-culture experiments and analyzed the same way as iPSC-Heps.

### Differentiation of iPSC-Macs
M1 and M0 iPSC-Macs were generated using a recently published protocol (M0 there referred to as M2)[39]. iPSC-Macs were seeded and polarized in ThinCert transwell inserts with 0.4 µm pore size (Greiner BioOne) 24 h before co-culture with iPSC-Heps.

### Isolation and differentiation of PHMs
Human peripheral blood mononuclear cells (PBMCs) from three healthy male donors were isolated by Biocoll (Sigma-Aldrich) density gradient centrifugation. PBMCs were washed and seeded in six-well plates at a density of $1 \times 10^6$ cells/cm² in Monocyte Attachment Medium (PromoCell). After attachment, monocytes were washed three times with PBS to remove non-adherent cells and differentiated into macrophages in ImmunoCult Macrophage Medium (StemCell Technologies) including 100 ng/ml MCSF for 7 days at 37 °C in 5% $CO_2$ and 5% $O_2$ with media exchange every 2 days. Purity of adherent monocytes was verified 24 h after isolation by flow cytometry analysis of CD14. After 7 days, PHMs were detached using Accutase, seeded and polarized in ThinCert transwell inserts with 0.4 µm pore size 24 h before co-culture with PHHs, as done for iPSC-Macs.

### Co-culture of iPSC-Heps/PHHs with iPSC-Macs/PHMs and cytokine neutralization
iPSC-Heps were cultured in 24-well plates without iPSC-Macs or with isogenic M1 or M0 iPSC-Macs at a ratio of 6:1 in a 50:50 mixture of HCM without insulin and hydrocortisone and RPMI including 10% fetal bovine serum (Gibco), 1% penicillin/streptomycin (Gibco) and 100 ng/ml MCSF (Peprotech). PHHs and PHMs were co-cultured the same way. For cytokine neutralization, antibodies against TNFα (Infliximab; Selleckchem) or IL1β (Human IL-1 beta/IL-1F2 Antibody; R&D Systems) were added to co-culture medium at 5 µg/ml or 0.2 µg/ml, respectively. The transwell inserts containing iPSC-Macs/PHMs were removed after 24 h of co-culture for subsequent analysis of media and cells and for hormonal stimulation of iPSC-Heps/PHHs.

### Hormonal stimulation of iPSC-Heps
iPSC-Heps were cultured without dexamethasone and insulin for 24 h at 37 °C in 5% $CO_2$ and 20% $O_2$. After washing twice with PBS, medium was changed to DMEM (Gibco) containing 2 mM sodium pyruvate (Gibco), 10 mM sodium lactate (Sigma-Aldrich) and 5.55 mM glucose (Gibco) unless stated otherwise in the figure legends. In addition, 100 nM glucagon (EMD Millipore) was added 1 h (except Fig. 1b, c and Supplementary Fig. 2a, b) before stimulation with 100 nM insulin. Hormonal stimulation of PHHs was done the same way.

### Oxygen consumption measurements
Mitochondrial respiration in the presence of electron transport chain inhibitors and uncouplers (oligomycin, 1.5 µM; carbonyl cyanide-p-trifluoromethoxyphenylhydrazone, 2 µM; rotenone, 0.5 µM; antimycin A, 0.5 µM) was measured in adherent iPSCs and iPSC-Heps using a Seahorse XFe24 Analyzer (Agilent) and the Mito Stress Test Kit (Agilent) according to the manufacturer's instructions. Oxygen consumption rates were normalized in each well by the number of cells assessed by nucleus counting after Hoechst 33342 (Sigma-Aldrich) staining using CellProfiler image analysis software.

### Fatty acid treatment of iPSC-Heps
iPSC-Heps were cultured without dexamethasone and insulin and treated every 48 h with 100 µM oleate (Sigma-Aldrich) and 100 µM palmitate (Sigma-Aldrich) for 6 days.

### Cytokine stimulation of iPSC-Heps
Fully differentiated iPSC-Heps were stimulated with either IL1β (10 ng/ml; Peprotech), TNFα (20 ng/ml; Peprotech), IL6 (20 ng/ml; Peprotech) or IFNγ (20 ng/ml; Peprotech) for 24 h in HCM without insulin and hydrocortisone before analysis.

### RNA isolation
RNA was isolated using PureLink RNA Mini Kit (Invitrogen) according to the manufacturer's instructions.

### qRT-PCR
cDNA was synthesized from purified RNA using qScript cDNA SuperMix (QuantaBio) according to the manufacturer's instructions. qRT-PCR was performed using SYBR Green PCR Master Mix (Applied Biosystems) in a ViiA 7 Real-Time PCR system (Applied Biosystems) using Quantstudio Real-Time PCR software (Applied Biosystems) for analysis. Oligonucleotide primers for each target gene were designed using the Primer3Plus website and synthesized by Integrated DNA Technologies. Relative mRNA expression was determined by the ΔΔ-Ct method normalized to *RPLPO*. Primers for qRT-PCR are listed in Supplementary Table 1.

### Immunofluorescence and triglyceride staining
Cells were washed twice with PBS and fixed with 4% paraformaldehyde (Thermo Fisher Scientific) in PBS for 10 min at room temperature. Afterwards, cells were washed three times and blocked/permeabilized in PBS containing 0.1% saponin (Sigma-Aldrich) and 3% normal donkey serum (NDS; Thermo Fisher Scientific) for 1 h at room temperature. Primary antibody incubation was performed at 4 °C overnight in PBS containing 0.1% saponin and 0.3% NDS. Afterwards, cells were washed three times with PBS containing 0.1% saponin and incubated for 1 h in PBS containing 0.1% saponin and 0.3% NDS with the respective secondary antibody. For triglyceride staining, permeabilized cells were incubated for 1 h with 2.5 µM BODIPY 493/503 (Cayman Chemicals) in PBS containing 0.1% saponin. Nuclei were stained with DAPI (Thermo Fisher Scientific). Cells were mounted in Fluoromount-G (Southern Biotech) before analysis. Images were acquired using an IX-71 microscope (Olympus). 4 random regions were analyzed per sample for quantification. Primary and secondary antibodies are listed in Supplementary Table 2.

### Mitochondria and viability staining
For mitochondria staining, iPSCs and iPSC-Heps were washed once with PBS and incubated in PBS containing Hoechst 33342 (341/486 nm) and MitoTracker Red FM (mitochondrial potential-dependent; Thermo Fisher Scientific) at 500 nM for 20 min at 37 °C and 5% $CO_2$ in the dark. Staining solution was then removed, cells were washed three times with PBS and immediately imaged using a BioTek Cytation cell imaging reader (BioTek). For viability staining, iPSC-Heps were washed once with PBS and incubated in PBS containing 1 µM Calcein-AM (Biolegend) and 1 µg/ml propidium iodide (Sigma-Aldrich) for 15 min. Staining solution was then removed, cells were washed twice with PBS and images were acquired using an Olympus IX-71 microscope.

### Triglyceride measurements
Triglycerides were measured using the Triglyceride Assay Kit (Biovision) according to the manufacturer's instructions. Briefly, iPSC-Heps were homogenized in 200 µl 5% NP-40 after which samples were slowly heated twice to 100 °C for 3 min followed by cooling to room temperature. Subsequently, samples were centrifuged at 16,000 × g,

supernatants were mixed with lipase for 20 min and then glycerol content was measured using a Synergy HT microplate reader (BioTek). Results were normalized to the total cellular protein content. Triglycerides were also measured using the Triglyceride-Glo assay (Promega) according to the manufacturer's instructions and analyzed using a Synergy HT microplate reader.

## Flow cytometry analysis

Freshly thawed PHHs, PHMs detached using Accutase, fully differentiated iPSC-Heps detached using 0.25% trypsin-EDTA and M1/M0 iPSC-Macs detached using Accutase were incubated with the respective antibodies in PBS including 0.1% BSA and 2 mM EDTA for 20 min at 4 °C and washed once before flow cytometry analysis. Antibodies are listed in Supplementary Table 2. Viable cells were distinguished using SYTOX Green/Blue Dead Cell Stain (Invitrogen). Unstained cells were used as control. Cells were analyzed using a LSRFortessa flow cytometer (BD Biosciences) and FloJo software (v10.6.1; BD Biosciences). A representative gating strategy example is shown in Supplementary Fig. 8.

## Cytokine analysis

Cytokine concentrations were measured in co-culture supernatants using LEGENDplex Human Inflammation Panel 1 (Biolegend) according to the manufacturer's instructions, analyzed using a LSRFortessa flow cytometer and quantified using LEGENDplex software (Biolegend). In addition, samples were sent to Eve Technologies for analysis with the Human Cytokine/Chemokine 48-Plex Discovery Assay (HD48); fluorescence intensity was used to generate cytokine heatmaps.

## Protein quantification and western blotting

All procedures were performed on ice or at 4 °C unless stated otherwise. Cells were lysed for 10 min in 1x RIPA buffer (Cell Signaling) including Halt Proteinase and Phosphatase Inhibitor Cocktail (Thermo Fisher Scientific). Lysates were centrifuged at $16,000 \times g$ for 20 min. Protein concentration was measured using Pierce BCA Protein Assay Kit (Thermo Fisher Scientific) kit according to the manufacturer's instructions. 4-15% pre-cast gels (Bio-Rad) were loaded with equal protein amounts for each individual experiment and run at 100 V for 75 min at room temperature. Protein was transferred to a nitrocellulose membrane (Bio-Rad) in transfer buffer at 100 V for 60 min. Membranes were blocked for 1 h at room temperature in TBS buffer containing 0.1% Tween 20 (TBST; Thermo Fisher Scientific) and 5% milk. Membranes were washed once in TBST and incubated in primary antibody solution (TBST containing 5% bovine serum albumin) overnight. Membranes were washed thrice in TBST and incubated for 1 h at room temperature in secondary antibody solution (TBST containing 5% bovine serum albumin). Chemiluminescence assays were performed using Pierce ECL Western Blotting Substrate (Thermo Fisher Scientific) and detected with a ChemiDoc XRS+ system (Bio-Rad). Image analyses and band density quantification were performed using Image Lab software (Bio-Rad) and FIJI open-source software. Quantification of relative protein phosphorylation changes was done separately for each biological replicate/individual blot. Primary and secondary antibodies are listed in Supplementary Table 2.

## Glucose measurement

Glucose in media was measured using Amplex Red Glucose/Glucoseoxidase Kit (Invitrogen) according to the manufacturer's instructions. Glucose concentration was normalized to total cellular protein.

## LDH measurement

LDH in media was measured using LDH Cytotoxicity Assay (ScienCell) according to the manufacturer's instructions.

## Glycogen staining and measurement

Glycogen staining was performed using Epredia's Perodic Acid Schiff Kit (Thermo Fisher Scientific) according to the manufacturer's instructions. Intracellular glycogen was quantified using the Glycogen-Glo assay (Promega) according to the manufacturer's instructions.

## RNA sequencing and bioinformatic analysis

mRNA library preparation with polyA enrichment and sequencing was performed by Novogene. Sequencing reads were aligned to the human reference genome GRCh38.96 and reads per gene matrix were counted with the latest Ensemble annotation build using STAR_2.7.2b[74]. Read counts per gene were used as input to DESeq2 v1.30.1[75] to determine differential gene expression between conditions using the Wald test while correcting for possible covariates. Genes passing a multiple test correcting $P < 0.05$ (FDR method) were considered significant. Data were analyzed by Gene Ontology analysis[76] and Kyoto Enrichment of Genes and Genomes pathway analysis[77] in DAVID and Ingenuity Pathway Analysis (IPA; QIAGEN). Venn diagram was generated using Venny v2.1.0 (open source); R software v4.0.2 (open source) was used for RNAseq data analysis. R software information and packages are listed in Supplementary Table 3.

## Statistics and reproducibility

Data were analyzed using Prism software (GraphPad) (except RNAseq data), including statistical analysis using one-way ANOVA (Dunnett's and Tukey's multiple comparison tests) and two-way ANOVA (Dunnett's multiple comparison test and two-stage step-up method of Benjamini, Krieger and Yekutieli). Due to the relatively small sample size normality testing was not feasible and all data were assumed to have a normal distribution. Group comparisons are indicated in the figure legends and $P$ values $< 0.05$ were considered statistically significant and indicated as $*P < 0.05$, $**P < 0.01$ and $***P < 0.001$. In whisker/box plots all data points are included, whiskers extend from minima to maxima, boxes extend from 25th to 75th percentile and lines in boxes represent median. $n$ values refer to biologically independent replicates of analyzed cells or media. All experiments were repeated independently with similar results at least three times, except for Supplementary Fig. 1e (flow cytometry analysis of freshly thawed PHHs). $P$ values of statistically significant results are provided in Supplementary Table 4.

## Reporting summary

Further information on research design is available in the Nature Portfolio Reporting Summary linked to this article.

# Data availability

Primer sequences, antibody sources, software information and $P$ values are provided in Supplementary Tables 1–4. Raw RNAseq data were deposited in the NCBI Gene Expression Omnibus database repository under accession number GSE228765. Processed RNAseq data are provided in Supplementary Data 1. Source data are provided with this paper.

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

## Acknowledgements

The authors thank Tania Moody and Walter Eckalbar for help with RNA-seq data analysis, Jae-Jun Kim for help with flow cytometry, Beltrán Borges and Simon N. Chu for donating and drawing blood, Suneil Koliwad for discussion and Simone Kurial and Pamela Derish for manuscript editing. This study was funded by NIH UG3/UH3 DK120004 to H.W. and E.C.H., NIH P30 DK026743 (UCSF Liver Center) to H.W. and J.J.M., a grant from the UCSF Academic Senate to J.J.M., a grant from the Weston Havens Foundation to J.J.M. and D.L.G., the Robert L. Kroc Chair III in Rheumatic and Connective Tissue Disease to E.C.H., an individual fellowship from the German Research Foundation (GR 5417/1-1) to M.G., an individual fellowship from NIH T32 DK060414 to E.H.A. and an individual fellowship from the Sao Paulo Research Foundation (2022/08157-5) to K.A.T.-S.

## Author contributions

M.G. conceived the study, performed experiments, analyzed the data, generated the figures, performed the statistical analyses and wrote the manuscript. K.M., E.H.A., A.P., D.L.G. and K.A.T.-S. performed experiments. C.P. analyzed the RNAseq data. E.C.H. and J.J.M. supervised experiments and edited the manuscript. H.W. conceived the study, supervised experiments and wrote the manuscript.

## Competing interests

The authors declare no competing interests.
