## [Peer Review File · Nature Communications]

Modeling and therapeutic targeting of inflammation-induced hepatic insulin resistance using human iPSC-derived hepatocytes and macrophagesREVIEWER COMMENTS

Reviewer #1 (Remarks to the Author):

Groeger et al first augment iPSC to hepatocyte differentiation protocol to improve insulin signaling in the resulting hepatocyte-like cells and then co-culture these cells with iPSC-derived macrophages in either the M1 or M2 states to determine the effects of such co-culture on insulin signaling. Cytokines secreted by the macrophages were identified via RNAseq and then inhibited these cytokines individually and in combination to elucidate ensuing effects on insulin signaling in the hepatocyte-like cells. In addition to improving insulin sensitivity in iPSC-derived hepatocyte-like cells, these studies showed that inflammation resulting from the macrophages is sufficient to induce insulin resistance (i.e., reduced effects of insulin on inhibition of gluconeogenesis and glycogenolysis and activation of glycolysis) in the hepatocyte-like cells. Overall, this study is well executed, the manuscript is well written, and the findings are indeed interesting and novel for the use of iPSC-derived hepatocyte-like cells for this space. The protocols developed for iPSC-derived hepatocyte-like cell maturation for better insulin signaling, co-culture with iPSC-derived macrophages in M1 and M2 states, the finding that inflammation can induce insulin resistance in hepatocyte-like cells even in the absence of hepatic steatosis, and identification of a combination of IL1b and TNFa as mediators of this response should be well received by academic and industrial investigators working in these areas. That being said, I have some suggestions below to make this manuscript more impactful to a broader audience.

1. A single iPSC line was used for all studies in the manuscript. Given how variable results can be from iPSC line to line, showing the major findings at least in another iPSC line would be important.

2. The authors point out the use of isogenic hepatocyte-like and macrophage-like cells in their studies. But, they don't take advantage of this aspect of the platform in modeling genotype-phenotype relationships. It would be useful to show that key mutations implicated in non-alcoholic fatty liver disease could be modeled in this co-culture platform. Reciprocally, if the authors had done experiments with hepatocyte-like cells from one iPSC line and macrophage-like cells from another line, would it have led to different results (for primary cells it often does not)?

3. The authors show that steatosis alone does not cause insulin resistance but they cite studies where steatosis (via de novo lipogenesis) in primary hepatocytes does cause insulin resistance. Even fatty acids and steatosis resulting from fat loading in primary hepatocytes can cause insulin resistance within a few days. While these previous findings do not take away from the authors' findings of inflammation-induced insulin resistance in hepatocyte-like cells, the authors should nonetheless reconcile what is known in primary hepatocytes and what they are finding in hepatocyte-like cells; could this be indicative of the need to further mature the hepatocyte-like cells?

4. Primary human hepatocytes and primary Kupffer macrophages are available from several vendors and hepatocytes can be kept insulin sensitive in several models including spheroids (see papers by Magnus Ingelman-Sundberg). Some experiments comparing the hepatocyte-like and macrophage-like responses to primary cell responses would help determine how mature the iPSC-derived cells are, which will help provide areas for future improvement and also an appraisal of whether the iPSC platform is ready for pharmaceutical development or further mechanistic studies. As it stands now, it is hard to fully decipher which findings are iPSC-specific and which would be replicated in functionally stabilized primary liver cells and at what quantitative level.

Reviewer #2 (Remarks to the Author):

In this paper by Groeger et al., authors generate hepatocytes and macrophages from human iPSC, and further co-cultivate iPSC-hepatocytes with condition medium of differentiated iPSC-macrophages. Glucose metabolism and inflammation associated aspects are measured in this model. Authors perform RNA sequencing analysis to reveal bulk transcriptomic alterations. Conclusively, authors indicate that M1 macrophage and its secretion of TNF α and IL1 β cause inflammation and insulin resistance in iPSC-Hepatocytes. Generally, this is an interesting approach to establish in-vitro hepatic glucose metabolism model with macrophage involvement. However, there are still several apparent disadvantages, which would very likely bring critical and controversial discussions.

Major comments:

1. Authors claim they generate iPSC-heps/-Macs following a published protocol. However, to validate this model (especially as to cells from artificial approaches), cell purity, viability and key features should be provided, before or during the further experiments.
2. Glucose production and relevant gene expression can suggest the glucose metabolism in hepatocytes. Nevertheless, the evidence is weak. More phenomenon and function experiments are required, such as the mitochondria function or key protein measurement.
3. Necessarily, the glucose production should be measured in iPSC-Macs. Even though authors show nice trends of glucose production in co-cultivated iPSC-Heps, disturbance of iPSC-Macs produced glucose would cause suspicious understanding.
4. In Figure 3, there are only 11 differentially up-regulated genes in condition medium treated iPSC-heps 'M2 vs. Ctrl'. In contrast to 'M1 vs. Ctrl' or 'M1 vs. M2', it appears that almost no alterations that M2 Macs bring to Heps, which is confusing. Considering no more evidence are provide for cell identification, the M2 differentiation is not so convincing.
5. More cell resources (such as primary human/mouse hepatocytes and macrophages) should be included to confirm or compare results from iPSC-hepatocytes /-Macs.
6. In Supplementary Fig. 3, authors claim a failure of cytokine stimulation (e.g. IL1b, TNF α) in iPSC-Heps. Considering the concentrations in Mac-condition medium are even lower, why do authors still believe that IL1b and TNF α serve as main mediator in hepatic glucose metabolism.

Minor comments:

1. In Figure 1d, the time duration (30 min) of experiments should be extended. And why are time points different between GYS/pGYS and others?
2. Methodologically, the quantification analysis from protein bands (WB) is not reliable and appropriate to indicate significant changes of protein levels. I would suggest authors not to display them in main results.
3. The efficiency of TNF- α and IL1b neutralization should be provided.
4. The Caspase 1 is important to cellular pyroptosis, but not enough for confirmation if only with gene expression elevation.
5. Other than cytokine neutralization, gene silencing or molecular inhibition should be applied on M1 Macs to compare and confirm the significance of IL1b and TNF α within the co-culture system.

Reviewer #3 (Remarks to the Author):

In this submission Groeger et al. outline a novel iPSC approach to studying hepatic insulin resistance and mechanisms involved. Specifically they sought to test whether inflammation or lipid accumulation/lipotoxicity are the primary drivers of hepatocyte insulin resistance. The experimental design was reasonable, outcomes appear to have been carefully and appropriately assessed, and conclusions largely supported by the experimental data. Exposing iPSC derived hepatocytes to pro-inflammatory iPSC derived macrophages, but not lipid exposure, induces the classic hallmarks of hepatocyte insulin resistance, including reduced insulin signal transduction, impaired regulation of glycolysis, glycogenolysis, gluconeogenesis, and glycogen synthesis as well as phenotypic excess production of glucose. Many of the experiments required conditions where baseline glucagon action was required to observe effects, however, this is also true in vivo, and thereby quite physiological. Cytokine neutralization experiments supported mechanistic findings as did the informatics approaches deployed. The manuscript is very well written. Of course, hepatic insulin resistance is of primary clinical importance and a study shedding light upon mechanisms would be of high importance and impact. Of many advantages of the iPSC approach is the ability to derive hepatocyte and macrophage models from multiple individuals, such as those with hepatic insulin sensitivity versus resistance. Additional advantages include the ability to manipulate the genome with crispr technology. Thus, my major apprehension is that this body of work was completed with a single iPSC cell line. It seems the work could be strengthened with validation of key findings in a second cell line, further validating mechanisms by genetically altering signaling through NF- κ B or JNK, or by generating cell lines from clinically relevant donors.

Minor concerns:

- 1) One domain of the findings employs changes in the expression of genes involved in hepatocyte glucose regulation. This is reasonable given that generally speaking there is some metabolic control associated with their expression level, however most of these enzymes also display significant allosteric regulation, which could be briefly recognized in the manuscript. For one example, GK has a regulatory partner in GKRP.
- 2) The figures are well designed and executed, but at review size are a bit of a challenge to read.

Point-by-point response to the reviewers' comments

Reviewer comments in black font, author responses in blue font (new or revised text in manuscript also in blue font)

Reviewer #1 (Remarks to the Author):

Groeger et al first augment iPSC to hepatocyte differentiation protocol to improve insulin signaling in the resulting hepatocyte-like cells and then co-culture these cells with iPSC-derived macrophages in either the M1 or M2 states to determine the effects of such co-culture on insulin signaling. Cytokines secreted by the macrophages were identified via RNAseq and then inhibited these cytokines individually and in combination to elucidate ensuing effects on insulin signaling in the hepatocyte-like cells. In addition to improving insulin sensitivity in iPSC-derived hepatocyte-like cells, these studies showed that inflammation resulting from the macrophages is sufficient to induce insulin resistance (i.e., reduced effects of insulin on inhibition of gluconeogenesis and glycogenolysis and activation of glycolysis) in the hepatocyte-like cells. Overall, this study is well executed, the manuscript is well written, and the findings are indeed interesting and novel for the use of iPSC-derived hepatocyte-like cells for this space. The protocols developed for iPSC-derived hepatocyte-like cell maturation for better insulin signaling, co-culture with iPSC-derived macrophages in M1 and M2 states, the finding that inflammation can induce insulin resistance in hepatocyte-like cells even in the absence of hepatic steatosis, and identification of a combination of IL1b and TNFa as mediators of this response should be well received by academic and industrial investigators working in these areas. That being said, I have some suggestions below to make this manuscript more impactful to a broader audience.

We appreciate the reviewer's feedback, both the recognition of the novelty and quality of our study and the insightful suggestions for maximizing its impact.

1. A single iPSC line was used for all studies in the manuscript. Given how variable results can be from iPSC line to line, showing the major findings at least in another iPSC line would be important.

We confirmed our main findings made in the widely used iPSC line WTC in another healthy-donor iPSC line CW10030, which was established as an effective control in a recent publication analyzing the transcriptome of iPSC-Heps generated from NAFLD patients¹. Our new data show that co-culture of CW10030 iPSC-Heps with isogenic M1 iPSC-Macs causes the same inflammatory phenotype in iPSC-Heps as the one identified in our original experiments using WTC iPSC line derivatives, including release of TNF α and IL1 β into the co-culture media as well as increased phosphorylation of JNK at T183/Y185 and increased gene expression of *NFKB2* and *TNF* in iPSC-Heps (new data in Fig. 2c, Supplementary Fig. 3e).

Our new data also show the same inflammation-induced insulin resistance phenotype in derivatives of the CW10030 and WTC iPSC lines, as evidenced by increased glucose production and decreased phosphorylation of AKT at S473, reduced dephosphorylation of PYGL at S15 as well as increased *PCK1* and decreased *GCK* gene expression in CW10030 iPSC-Heps co-cultured with isogenic M1 iPSC-Macs; importantly, we also found that antibody-mediated neutralization of TNF α and IL1 β is as effective in restoring insulin sensitivity in CW10030 iPSC-Heps co-cultured with isogenic M1 iPSC-Macs as it is in WTC iPSC line derivatives (new data in Fig. 5d-f, Supplementary Fig. 7c).

2. The authors point out the use of isogenic hepatocyte-like and macrophage-like cells in their studies. But, they don't take advantage of this aspect of the platform in modeling genotype-phenotype relationships. It would be useful to show that key mutations implicated in non-alcoholic fatty liver disease could be modeled in this co-culture platform. Reciprocally, if the authors had done experiments with hepatocyte-like cells from one iPSC line and macrophage-like cells from another line, would it have led to different results (for primary cells it often does not)?

Our main motivation for establishing co-culture of iPSC-Heps with isogenic iPSC-Macs was to exclude bias from our mechanistic studies of how inflammation impacts the complex regulation of glucose metabolism by insulin. In doing so, we sought to develop an experimental system that provides the authenticity and robustness needed for successful development of therapeutics for hepatic insulin resistance. We very much appreciate the reviewer's ideas of taking advantage of these characteristics of our model to investigate genotype-phenotype relationships and have added this point to the discussion of our revised manuscript (line 348).

3. The authors show that steatosis alone does not cause insulin resistance but they cite studies where steatosis (via de novo lipogenesis) in primary hepatocytes does cause insulin resistance. Even fatty acids and steatosis resulting from fat loading in primary hepatocytes can cause insulin resistance within a few days. While these previous findings do not take away from the authors' findings of inflammation-induced insulin resistance in hepatocyte-like cells, the authors should nonetheless reconcile what is known in primary hepatocytes and what they are finding in hepatocyte-like cells; could this be indicative of the need to further mature the hepatocyte-like cells?

We agree with the reviewer's point that steatosis has been reported to cause insulin resistance in primary hepatocytes and have revised our introduction accordingly, including referencing the paper highlighted in their comment 4 below (line 74)². We also substantiated our original finding that treating iPSC-Heps with the fatty acids oleate and palmitate (100 μ M each) every 48 hours for 6 days does not cause insulin resistance. For this, we added new data from comparison of iPSC-Heps generated from 3 iPSC line donors homozygous for the PNPLA3 I148M variant who have biopsy-confirmed NASH and 3 healthy-donor iPSC lines¹. As expected and previously reported³, the PNPLA3 I148M iPSC-Heps spontaneously accumulated triglycerides; however, despite steatosis, the cells maintained normal insulin sensitivity as evidenced by unaltered glucose output and *PCK1* and *SREBP1c* gene expression after insulin bolus (new data in Fig. 2h-j, Supplementary Fig. 4m). These new data accord with the clinical observation that PNPLA3 I148M-associated steatosis of hepatocytes is not linked to hepatic insulin resistance^{4,5}, which supports the authenticity of our iPSC-Heps.

A potential explanation for previous findings of fatty-acid-induced insulin resistance not being detectable in our iPSC-Heps is that we used lower (non-lipotoxic⁶) fatty acid doses or did not include other metabolic challenges such as high insulin or high glucose². Along these lines, it is possible that steatosis-induced hepatic insulin resistance depends on intracellular lipid composition⁴ and/or manifestation of lipotoxicity and ER stress, which act through some of the same inflammatory signaling pathways, particularly JNK, that we identified to mediate induction of hepatic insulin resistance by macrophage-mediated inflammation⁷⁻⁹. In accord with this potential explanation, which we discuss in our revised manuscript (lines 317-319), we found unaltered gene expression of *JUN*, a transcription factor activated by JNK under lipotoxic conditions¹⁰, in steatotic PNPLA3 I148M iPSC-Heps (new data in Supplementary Fig. 4n).

4. Primary human hepatocytes and primary Kupffer macrophages are available from several vendors and hepatocytes can be kept insulin sensitive in several models including spheroids (see papers by Magnus Ingelman-Sundberg). Some experiments comparing the hepatocyte-like and macrophage-like responses to primary cell responses would help determine how mature the iPSC-derived cells are, which will help provide areas for future improvement and also an appraisal of whether the iPSC platform is ready for pharmaceutical development or further mechanistic studies. As it stands now, it is hard to fully decipher which findings are iPSC-specific and which would be replicated in functionally stabilized primary liver cells and at what quantitative level.

We thank the reviewer for the suggestion to illustrate the authenticity and utility of our iPSC-based model by validating it using co-cultures of primary human hepatocytes (PHHs) and primary human macrophages (PHMs). Our new data replicate in primary cells our finding of inflammation-induced hepatic insulin resistance made in iPSC derivatives, including the underlying mechanisms and mediators.

Specifically, our new data show that co-culture with M1 PHMs causes the same inflammatory phenotype in PHHs as identified in our experiments using WTC or CW10030 iPSC line derivatives, including release of $TNF\alpha$ and $IL1\beta$ into the co-culture media and increased gene expression of *NFKB2*, *TNF* and *CASP1* in PHHs (new data in Fig. 5a, c). We also found increased phosphorylation of JNK at T183/Y185 in PHHs, which occurred already after 2 hours of co-culture with M1 PHMs and was not sustained until 24 hours as in iPSC-Heps (new data in Fig. 5b, Supplementary Fig. 7a). However, gene expression of *JUN*, the downstream effector of JNK-mediated inflammation¹⁰, remained increased at 24 hours and, exactly as in iPSC-Heps co-cultured with M1 iPSC-Macs, antibody-mediated neutralization of $TNF\alpha$ and $IL1\beta$ was effective in reducing *JUN*, *NFKB2* and *TNF* but not *CASP1* gene expression in PHHs co-cultured with PHMs (new data in Fig. 5c).

Our new data also show the inflammation-induced insulin resistance phenotype in PHHs co-cultured with M1 PHMs that we identified using derivatives of the WTC and CW10030 iPSC lines, as evidenced by increased glucose production and decreased phosphorylation of AKT at S473 as well as increased *PCK1*, decreased *GCK* and increased *G6PC* gene expression in PHHs (new data in Fig. 5d-f, Supplementary Fig. 7b). We did not find reduced dephosphorylation of PYGL at S15, which we attribute to the much lower glycogen levels in PHHs than in iPSC-Heps causing suppression of glycogenolysis as discussed in our revised manuscript (new data in Supplementary Figs. 7c, 2e; lines 326-329). Importantly, we found that antibody-mediated neutralization of $TNF\alpha$ and $IL1\beta$ is as effective in restoring insulin sensitivity in PHHs co-cultured with M1 PHMs as it is in derivatives of the WTC and CW10030 iPSC lines (new data in Fig. 5d-f, Supplementary Fig. 7b).

Reviewer #2 (Remarks to the Author):

In this paper by Groeger et al., authors generate hepatocytes and macrophages from human iPSC, and further co-cultivate iPSC-hepatocytes with condition medium of differentiated iPSC-macrophages. Glucose metabolism and inflammation associated aspects are measured in this model. Authors perform RNA sequencing analysis to reveal bulk transcriptomic alterations. Conclusively, authors indicate that M1 macrophage and its secretion of TNF α and IL1 β cause inflammation and insulin resistance in iPSC-Hepatocytes. Generally, this is an interesting approach to establish in-vitro hepatic glucose metabolism model with macrophage involvement. However, there are still several apparent disadvantages, which would very likely bring critical and controversial discussions.

We appreciate the reviewer's feedback, both the recognition of the originality of our study and the insightful suggestions for eliminating uncertainties from our approach.

Major comments:

1. Authors claim they generate iPSC-heps/-Macs following a published protocol. However, to validate this model (especially as to cells from artificial approaches), cell purity, viability and key features should be provided, before or during the further experiments.

We added new data validating the suitability of our iPSC derivatives for liver disease modeling to our revised manuscript, including analyses of viability (new data in Supplementary Fig. 1b) and lineage-specific differentiation and purity (new data in Supplementary Fig. 1c-e) of iPSC-Heps as well as analyses of viability (new data in Supplementary Fig. 3a) and lineage-specific differentiation and purity (new data in Supplementary Fig. 3b) of iPSC-Macs.

2. Glucose production and relevant gene expression can suggest the glucose metabolism in hepatocytes. Nevertheless, the evidence is weak. More phenomenon and function experiments are required, such as the mitochondria function or key protein measurement.

We view glucose production as strong and important evidence for glucose metabolism in hepatocytes because it is a direct and clinically relevant parameter that is lacking in many published experimental studies. However, we agree that additional data would substantiate our results and have added new data on protein phosphorylation in iPSC-Heps of the insulin receptor (new data in Supplementary Fig. 2c), the mTORC1 effector p70 S6 kinase (new data in Fig. 1d), which is essential for hepatic glucose homeostasis¹¹, and glycogen synthase kinase-3 β (new data in Supplementary Fig. 2d), data on mitochondrial viability and respiration in iPSC-Heps (new data in Supplementary Fig. 1f, g) and data on glycogen accumulation in iPSC-Heps to our revised manuscript (new data in Fig. 1e, Supplementary Fig. 2e).

3. Necessarily, the glucose production should be measured in iPSC-Macs. Even though authors show nice trends of glucose production in co-cultivated iPSC-Heps, disturbance of iPSC-Macs produced glucose would cause suspicious understanding.

The reviewer raises an interesting question about glucose production by iPSC-Macs, which, however, does not apply to our co-culture experiments because iPSC-Macs were separated from iPSC-Heps before analysis of insulin effects. Briefly, our co-culture approach employs transwell inserts with a pore size of 0.4 μ m that do not allow transmigration of iPSC-Macs. Before measuring insulin effects, we removed the transwells containing the iPSC-Macs and added new media with defined glucose content to the iPSC-Heps. This approach is described in the methods section (lines 417-419) and media glucose content is given in the figure legends of our manuscript.

Although our approach excludes a contribution of iPSC-Mac to measurements of glucose output by iPSC-Heps in our manuscript, we attempted to measure glucose production by iPSC-Macs. However, we could not detect measurable glucose production by iPSC-Macs in glucose-containing media and the cells were not sufficiently viable in glucose-free media (data not shown).

4. In Figure 3, there are only 11 differentially up-regulated genes in condition medium treated iPSC-heps 'M2 vs. Ctrl'. In contrast to 'M1 vs. Ctrl' or 'M1 vs. M2', it appears that almost no alterations that M2 Macs bring to Heps, which is confusing. Considering no more evidence are provide for cell identification, the M2 differentiation is not so convincing.

We agree with the reviewer that describing undifferentiated/non-activated macrophages as M2 was not accurate in our original manuscript. We adopted this term from the original publication of the iPSC-Mac differentiation protocol by our collaborators¹². However, that protocol does not use IL4 or other factors to generate iPSC-Macs that exhibit M2 functions such as promotion of regeneration¹³. In our revised manuscript we changed the designation from M2 to M0 to reflect the undifferentiated/non-activated state of these iPSC-Macs, as evidenced by low/absent secretion of cytokines/chemokines (Supplementary Fig. 3f), which explains the small number of differentially expressed genes between iPSC-Heps co-cultured with M0 iPSC-Macs and iPSC-Heps in mono-culture, and supports the use of M0 iPSC-Macs as a control for M1 iPSC-Macs. We added a statement about using the designation M0 instead of M2 (as used in the original published protocol)¹² to the methods section of our revised manuscript (lines 397-398).

5. More cell resources (such as primary human/mouse hepatocytes and macrophages) should be included to confirm or compare results from iPSC-hepatocytes /-Macs.

We thank the reviewer for the suggestion to illustrate the authenticity and utility of our iPSC-based model by validating it using co-cultures of primary human hepatocytes (PHHs) and primary human macrophages (PHMs). Our new data replicate in primary cells our finding of inflammation-induced hepatic insulin resistance made in iPSC derivatives, including the underlying mechanisms and mediators. In addition, we included new data that replicates our original findings made in the widely used iPSC line WTC in another healthy-donor iPSC line CW10030, which was established as an effective control in a recent publication analyzing the transcriptome of iPSC-Heps generated from NAFLD patients¹.

Specifically, our new data show that co-culture with M1 PHMs causes the same inflammatory phenotype in PHHs as identified in our experiments using WTC or CW10030 iPSC line derivatives, including release of TNF α and IL1 β into the co-culture media and increased gene expression of *NFKB2*, *TNF* and *CASP1* in PHHs (new data in Fig. 5a, c). We also found increased phosphorylation of JNK at T183/Y185 in PHHs, which occurred already after 2 hours of co-culture with M1 PHMs and was not sustained until 24 hours as in iPSC-Heps (new data in Fig. 5b, Supplementary Fig. 7a). However, gene expression of *JUN*, the downstream effector of JNK-mediated inflammation¹⁰, remained increased at 24 hours and, exactly as in iPSC-Heps co-cultured with M1 iPSC-Macs, antibody-mediated neutralization of TNF α and IL1 β was effective in reducing *JUN*, *NFKB2* and *TNF* but not *CASP1* gene expression in PHHs co-cultured with PHMs (new data in Fig. 5c).

Our new data also show the inflammation-induced insulin resistance phenotype in PHHs co-cultured with M1 PHMs that we identified using derivatives of the WTC and CW10030 iPSC lines, as evidenced by increased glucose production and decreased phosphorylation of AKT at S473 as well as increased *PCK1*, decreased *GCK* and increased *G6PC* gene expression in PHHs (new

data in Fig. 5d-f, Supplementary Fig. 7b). We did not find reduced dephosphorylation of PYGL at S15, which we attribute to much lower glycogen levels in PHHs than in iPSC-Heps causing suppression of glycogenolysis as discussed in our revised manuscript (Supplementary Figs. 7c, 2e; lines 326-329). Importantly, we found that antibody-mediated neutralization of TNF α and IL1 β is as effective in restoring insulin sensitivity in PHHs co-cultured with M1 PHMs as in derivatives of the WTC and CW10030 iPSC lines (new data in Fig. 5d-f, Supplementary Fig. 7b).

6. In Supplementary Fig. 3, authors claim a failure of cytokine stimulation (e.g. IL1b, TNFa) in iPSC-Heps. Considering the concentrations in Mac-condition medium are even lower, why do authors still believe that IL1b and TNFa serve as main mediator in hepatic glucose metabolism.

The concentrations used for the single-cytokine challenges shown in Supplementary Fig. 3a, b (Supplementary Fig. 4a, b in revised manuscript) are derived from previous publications¹⁴⁻¹⁷. We believe that this approach of applying cytokines individually, even at high doses, was not effective in our model because it is unphysiological: in vivo, cytokines do not act alone but in concert, as reflected by the secretory profile of our M1 iPSC-Macs (Supplementary Fig. 3f). Activation of overlapping inflammatory signaling pathways by different cytokines can be expected to amplify the cellular response and promote disease progression as we observed in co-cultures of iPSC-Heps with M1 iPSC-Macs. In addition, continuous cytokine secretion by M1 iPSC-Macs over 24-hour co-culture causes more profound activation of inflammatory signaling pathways in iPSC-Heps than boluses of cytokines added to the media because the half-life of cytokines such as TNF α and IL1 β is less than 30 minutes¹⁸.

In addition, we confirmed our RNAseq and cytokine profiling-based finding of TNF α and IL1 β being the main mediators of M1 iPSC-Mac-induced insulin resistance in iPSC-Heps using specific neutralizing antibodies in our original manuscript (Fig. 4d-g), which we further confirmed in co-cultures of derivatives of another iPSC line (CW10030 in addition to WTC) and co-cultures of primary human cells in our revised manuscript (new data in Fig. 5d-f, Supplementary Fig. 7b).

Minor comments:

1. In Figure 1d, the time duration (30 min) of experiments should be extended. And why are time points different between GYS/pGYS and others?

We analyzed phosphorylation of GYS at S641 at 8 time points over the course of 3 hours because we did not find significant changes during the 5-stage 30-minute time course that we used for analysis of phosphorylation of the other proteins involved in insulin signaling. Prompted by the reviewer's comment, we repeated this analysis and obtained the same result (new data in Fig. 1d). We expanded this analysis by investigating inactivating phosphorylation at S9 of glycogen synthase kinase-3 β , which is a negative regulator of GYS, over the same time period (new data in Supplementary Fig. 2d). We found a trend toward inactivation of glycogen synthase kinase-3 β , which, however, did not translate into GYS activation, prompting us to conclude that glycogen synthesis is suppressed in iPSC-Heps because they have very high levels of glycogen at baseline (Supplementary Fig. 2e). We added this point to the revised manuscript (lines 128-132).

We analyzed phosphorylation of the other proteins involved in insulin signaling at 5 time points over the course of 30 minutes because a similar approach has been used in analyses of primary hepatocytes referenced in our manuscript¹⁹. The rationale for a 30-minute time course is that non-insulin-related changes and/or feedback effects occurring at later time points could bias the results.

2. Methodologically, the quantification analysis from protein bands (WB) is not reliable and appropriate to indicate significant changes of protein levels. I would suggest authors not to display them in main results.

We respectfully disagree with the reviewer's comment as we have taken a well-controlled approach to western blot quantification. Briefly, we measured protein content before each experiment and used reference proteins as loading control to standardize our experimental approach. For each biological replicate, we quantified each gel/blot separately measuring density of protein bands as area under the curve. We normalized values for each band on a gel/blot to a reference condition for each biological replicate and displayed them as relative changes in protein phosphorylation. For detection of bands we used a dedicated automated Bio-Rad ChemiDoc XRS+ system to avoid appearance and quantification of saturated bands. We will provide all uncropped western blot images used for quantification in the source data.

3. The efficiency of TNF- α and IL1 β neutralization should be provided.

We purchased neutralizing antibodies against TNF α (Infliximab) and IL1 β (Human IL-1 beta/IL-1F2 Antibody) from Selleckchem and R&D Systems, respectively. The published neutralization efficiency of Infliximab is an effective neutralization of 1 ng TNF α by 100 ng/ml antibody²⁰. The neutralization efficiency for Human IL-1 beta/IL-1F2 Antibody given on the vendor's website (https://www.rndsystems.com/products/human-il-1beta-il-1f2-antibody-8516_mab201) is an effective neutralization (ED50) dose of 50 pg/ml IL1 β by 1-3 ng/ml antibody. We used an Infliximab dose of 5 μ g/ml at a measured TNF α concentration of less than 20 pg/ml in 0.7 ml media. As previously reported²¹, the Infliximab dose was effective in reducing CCR2 cell surface levels in M1 iPSC-Macs (new data in Supplementary Fig. 6b). We used a Human IL-1 beta/IL-1F2 Antibody dose of 0.2 μ g/ml with a measured IL1 β concentration of less than 25 pg/ml in 0.7 ml media. We used both antibodies at higher doses than the reported neutralization efficiencies to ascertain efficacy over the 24-hour co-culture period. Of note, our antibody doses are 50-200-fold lower than doses used in published studies^{22,23}.

4. The Caspase 1 is important to cellular pyroptosis, but not enough for confirmation if only with gene expression elevation.

We agree with the reviewer's point that *CASP1* gene expression is not sufficient to claim pyroptosis as the mechanism of cytokine-induced cell death. We substantiated this result by adding gene expression analysis of *CASP4* and *CASP5*²⁴ (new data in Supplementary Fig. 3d) and revised the text to indicate that other cell-death mechanisms might be involved (line 304).

5. Other than cytokine neutralization, gene silencing or molecular inhibition should be applied on M1 Macs to compare and confirm the significance of IL1 β and TNF α within the co-culture system.

We very much appreciate the reviewer's ideas for using our iPSC-based system for further mechanistic studies and have added this point to the discussion of our revised manuscript (line 348).

Reviewer #3 (Remarks to the Author):

In this submission Groeger et al. outline a novel iPSC approach to studying hepatic insulin resistance and mechanisms involved. Specifically they sought to test whether inflammation or lipid accumulation/lipotoxicity are the primary drivers of hepatocyte insulin resistance. The experimental design was reasonable, outcomes appear to have been carefully and appropriately assessed, and conclusions largely supported by the experimental data. Exposing iPSC derived hepatocytes to pro-inflammatory iPSC derived macrophages, but not lipid exposure, induces the classic hallmarks of hepatocyte insulin resistance, including reduced insulin signal transduction, impaired regulation of glycolysis, glycogenolysis, gluconeogenesis, and glycogen synthesis as well as phenotypic excess production of glucose. Many of the experiments required conditions where baseline glucagon action was required to observe effects, however, this is also true in vivo, and thereby quite physiological. Cytokine neutralization experiments supported mechanistic findings as did the informatics approaches deployed. The manuscript is very well written. Of course, hepatic insulin resistance is of primary clinical importance and a study shedding light upon mechanisms would be of high importance and impact. Of many advantages of the iPSC approach is the ability to derive hepatocyte and macrophage models from multiple individuals, such as those with hepatic insulin sensitivity versus resistance. Additional advantages include the ability to manipulate the genome with crispr technology. Thus, my major apprehension is that this body of work was completed with a single iPSC cell line. It seems the work could be strengthened with validation of key findings in a second cell line, further validating mechanisms by genetically altering signaling through NF-kB or JNK, or by generating cell lines from clinically relevant donors.

We appreciate the reviewer's feedback, both the recognition of the quality and clinical relevance of our study and the insightful suggestions for maximizing its impact.

As suggested by the reviewer, we confirmed our main findings made in the widely used iPSC line WTC in another healthy-donor iPSC line CW10030, which was established as an effective control in a recent publication analyzing the transcriptome of iPSC-Heps generated from NAFLD patients¹. Our new data show that co-culture of CW10030 iPSC-Heps with isogenic M1 iPSC-Macs causes the same inflammatory phenotype in iPSC-Heps as identified in our original experiments using WTC iPSC line derivatives, including release of TNF α and IL1 β into the co-culture media as well as increased phosphorylation of JNK at T183/Y185 and increased gene expression of *NFKB2* and *TNF* in iPSC-Heps (new data in Fig. 2c, Supplementary Fig. 3e).

Our new data also show the same inflammation-induced insulin resistance phenotype in derivatives of the CW10030 and WTC iPSC lines, as evidenced by increased glucose production and decreased phosphorylation of AKT at S473, reduced dephosphorylation of PYGL at S15 as well as increased *PCK1* and decreased *GCK* gene expression in CW10030 iPSC-Heps co-cultured with isogenic M1 iPSC-Macs; importantly, we also found that antibody-mediated neutralization of TNF α and IL1 β is as effective in restoring insulin sensitivity in CW10030 iPSC-Heps co-cultured with isogenic M1 iPSC-Macs as in WTC iPSC line derivatives (new data in Fig. 5d-f, Supplementary Fig. 7c).

As further suggested by the reviewer, we added new data from comparison of iPSC-Heps generated from 3 iPSC line donors homozygous for the PNPLA3 I148M variant who have biopsy-confirmed NASH and 3 healthy-donor iPSC lines¹ to substantiate our original finding that treating iPSC-Heps with fatty acids for 6 days does not cause insulin resistance. As expected and previously reported³, the PNPLA3 I148M iPSC-Heps spontaneously accumulated triglycerides; however, despite steatosis, they maintained normal insulin sensitivity as evidenced by unaltered

glucose output and *PCK1* and *SREBP1c* gene expression after insulin bolus (new data in Fig. 2h-j, Supplementary Fig. 4m). Our new data accord with the clinical observation that PNPLA3 I148M-associated steatosis of hepatocytes is not linked to hepatic insulin resistance^{4,5}, which supports the authenticity of our iPSC-Heps. A potential explanation for previous findings of fatty-acid-induced insulin resistance in primary hepatocytes not being detectable in our iPSC-Heps is that we used lower (non-lipotoxic⁶) fatty acid doses and did not include other metabolic challenges such as high insulin or high glucose². Moreover, it is possible that steatosis-induced hepatic insulin resistance depends on the intracellular lipid composition⁴ and/or manifestation of lipotoxicity and ER stress, which act through some of the same inflammatory signaling pathways, particularly JNK, that we identified to mediate induction of hepatic insulin resistance by macrophage-mediated inflammation⁷⁻⁹. In accord with this potential explanation, which we discuss in our revised manuscript (lines 317-319), we found unaltered gene expression of *JUN*, a transcription factor activated by JNK under lipotoxic conditions¹⁰, in steatotic PNPLA3 I148M iPSC-Heps (new data in Supplementary Fig. 4n).

We very much appreciate the reviewer's ideas for using our iPSC-based system for further mechanistic studies using gene editing and have added this point to the discussion of our revised manuscript (line 348).

Minor concerns:

1) One domain of the findings employs changes in the expression of genes involved in hepatocyte glucose regulation. This is reasonable given that generally speaking there is some metabolic control associated with their expression level, however most of these enzymes also display significant allosteric regulation, which could be briefly recognized in the manuscript. For one example, GK has a regulatory partner in GKRP.

We agree with the reviewer that allosteric regulation of enzymes plays an important role in hepatic glucose metabolism. We have highlighted the need for considering this additional level of regulation in future studies in the discussion of our revised manuscript (line 326-329).

2) The figures are well designed and executed, but at review size are a bit of a challenge to read.

We appreciate the reviewer's comment as we take pride in providing clear and clean figures. We are including larger versions in our revised manuscript.

References

1. Duwaerts, C.C. et al. Induced Pluripotent Stem Cell-derived Hepatocytes From Patients With Nonalcoholic Fatty Liver Disease Display a Disease-specific Gene Expression Profile. *Gastroenterology* **160**, 2591-2594.e6 (2021).
2. Kozyra, M. et al. Human hepatic 3D spheroids as a model for steatosis and insulin resistance. *Sci Rep* **8**, 14297 (2018).
3. Tilson, S.G. et al. Modeling PNPLA3-Associated NAFLD Using Human-Induced Pluripotent Stem Cells. *Hepatology* **74**, 2998–3017 (2021).
4. Franko, A. et al. Dissociation of Fatty Liver and Insulin Resistance in I148M PNPLA3 Carriers: Differences in Diacylglycerol (DAG) FA18:1 Lipid Species as a Possible Explanation. *Nutrients* **10** (2018).
5. Liu, Z. et al. Causal relationships between NAFLD, T2D and obesity have implications for disease subphenotyping. *J Hepatol* **73**, 263–276 (2020).
6. Zeng, X. et al. Oleic acid ameliorates palmitic acid induced hepatocellular lipotoxicity by inhibition of ER stress and pyroptosis. *Nutr Metab (Lond)* **17**, 11 (2020).
7. Chen, L., Chen, R., Wang, H. & Liang, F. Mechanisms Linking Inflammation to Insulin Resistance. *Int J Endocrinol* **2015**, 508409 (2015).
8. Mota, M., Banini, B.A., Cazanave, S.C. & Sanyal, A.J. Molecular mechanisms of lipotoxicity and glucotoxicity in nonalcoholic fatty liver disease. *Metabolism* **65**, 1049–1061 (2016).
9. Ozcan, U. et al. Endoplasmic reticulum stress links obesity, insulin action, and type 2 diabetes. *Science* **306**, 457–461 (2004).
10. Yung, J.H.M. & Giacca, A. Role of c-Jun N-terminal Kinase (JNK) in Obesity and Type 2 Diabetes. *Cells* **9** (2020).
11. Um, S.H., D'Alessio, D. & Thomas, G. Nutrient overload, insulin resistance, and ribosomal protein S6 kinase 1, S6K1. *Cell Metab* **3**, 393–402 (2006).
12. Matsuo, K. et al. ACVR1R206H extends inflammatory responses in human induced pluripotent stem cell-derived macrophages. *Bone* **153**, 116129 (2021).
13. Orekhov, A.N. et al. Monocyte differentiation and macrophage polarization. *VP* **2019** (2019).
14. Hirosumi, J. et al. A central role for JNK in obesity and insulin resistance. *Nature* **420**, 333–336 (2002).
15. Nov, O. et al. Interleukin-1beta may mediate insulin resistance in liver-derived cells in response to adipocyte inflammation. *Endocrinology* **151**, 4247–4256 (2010).
16. Kano, A., Watanabe, Y., Takeda, N., Aizawa, S. & Akaike, T. Analysis of IFN-gamma-induced cell cycle arrest and cell death in hepatocytes. *J Biochem* **121**, 677–683 (1997).
17. Senn, J.J., Klover, P.J., Nowak, I.A. & Mooney, R.A. Interleukin-6 induces cellular insulin resistance in hepatocytes. *Diabetes* **51**, 3391–3399 (2002).
18. Liu, C. et al. Cytokines: From Clinical Significance to Quantification. *Adv Sci (Weinh)* **8**, e2004433 (2021).

19. Molinaro, A., Becattini, B. & Solinas, G. Insulin signaling and glucose metabolism in different hepatoma cell lines deviate from hepatocyte physiology toward a convergent aberrant phenotype. *Sci Rep* **10**, 12031 (2020).
20. Buurman, D.J. et al. Quantitative comparison of the neutralizing capacity, immunogenicity and cross-reactivity of anti-TNF- α biologicals and an Infliximab-biosimilar. *PLoS One* **13**, e0208922 (2018).
21. Xia, L., Lu, J. & Xiao, W. Blockage of TNF- α by infliximab reduces CCL2 and CCR2 levels in patients with rheumatoid arthritis. *J Investig Med* **59**, 961–963 (2011).
22. Acosta-Rodriguez, E.V., Napolitani, G., Lanzavecchia, A. & Sallusto, F. Interleukins 1beta and 6 but not transforming growth factor-beta are essential for the differentiation of interleukin 17-producing human T helper cells. *Nat Immunol* **8**, 942–949 (2007).
23. Moriconi, F. et al. The anti-TNF- α antibody infliximab indirectly regulates PECAM-1 gene expression in two models of in vitro blood cell activation. *Lab Invest* **92**, 166–177 (2012).
24. Yu, P. et al. Pyroptosis: mechanisms and diseases. *Signal Transduct Target Ther* **6** (2021).

REVIEWERS' COMMENTS

Reviewer #1 (Remarks to the Author):

The authors have done a thorough job addressing my comments, including data from a second iPSC line and data from primary cells for comparison, as well as textual changes to better contextualize their findings relative to the existing literature. The manuscript is novel and impactful and is ready for publication.

Reviewer #2 (Remarks to the Author):

Thanks a lot!

I appreciate all efforts that authors have made. Responses and revisions in the manuscript addressed the initial questions very well.

Although some minor discussions may still exist, I would recommend editors consider a final acceptance.

Sincerely looking forward to this paper being smoothly published.

Best regards,

Dr. MD. Hanyang Liu

Reviewer #3 (Remarks to the Author):

The authors, in my view, have been quite responsive to the original review, have confirmed findings in additional iPSCs and in primary cells. While this model certainly has caveats in understanding in vivo complex pathophysiology, they have made a compelling case that this model is worth further consideration by the field.